# Assessment of Inner Blood–Retinal Barrier: Animal Models and Methods

**DOI:** 10.3390/cells12202443

**Published:** 2023-10-12

**Authors:** Kiran Bora, Neetu Kushwah, Meenakshi Maurya, Madeline C. Pavlovich, Zhongxiao Wang, Jing Chen

**Affiliations:** Department of Ophthalmology, Boston Children’s Hospital, Harvard Medical School, 300 Longwood Avenue, Boston, MA 02115, USA

**Keywords:** blood–retinal barrier, vascular leakage, Wnt/β-catenin signaling, animal models, leakage assays

## Abstract

Proper functioning of the neural retina relies on the unique retinal environment regulated by the blood–retinal barrier (BRB), which restricts the passage of solutes, fluids, and toxic substances. BRB impairment occurs in many retinal vascular diseases and the breakdown of BRB significantly contributes to disease pathology. Understanding the different molecular constituents and signaling pathways involved in BRB development and maintenance is therefore crucial in developing treatment modalities. This review summarizes the major molecular signaling pathways involved in inner BRB (iBRB) formation and maintenance, and representative animal models of eye diseases with retinal vascular leakage. Studies on Wnt/β-catenin signaling are highlighted, which is critical for retinal and brain vascular angiogenesis and barriergenesis. Moreover, multiple in vivo and in vitro methods for the detection and analysis of vascular leakage are described, along with their advantages and limitations. These pre-clinical animal models and methods for assessing iBRB provide valuable experimental tools in delineating the molecular mechanisms of retinal vascular diseases and evaluating therapeutic drugs.

## 1. Introduction

The eye, as a highly specialized sensory organ and part of the central nervous system (CNS), requires an adequate supply of oxygen and nutrients with restricted passage of harmful molecules, to maintain an ideal neural tissue environment for proper visual function. The neurosensory retina is one of the most metabolically active tissues in our body per tissue weight, with the light-sensing photoreceptor cells being the major oxygen consumers [1,2]. In vascularized retinas in most mammals including both human and rodents, the high oxygen demands are met by a complex and well organized vascular system consisting of the retinal and choroidal circulation, which nourishes the inner and outer retina, respectively [3].

Along with surplus oxygen supply, efficient functioning of the neural retina requires strictly controlled flow of ions, water, amino acids, metabolic waste products, and sugar, as well as the exclusion of antibodies, immune cells, and pathogens from the circulation. This is mediated by the blood–retinal barrier (BRB), comprised of both inner BRB (iBRB) and outer BRB (oBRB) [4]. The inner BRB contains mostly a vascular component, analogous to the blood–brain barrier (BBB), established by the retinal capillary endothelium tight junctions (TJs) and end feet of surrounding mural cells including pericytes and astrocytes. The iBRB forms the blood–neural barrier in the inner retina, regulating the exchange of solutes between the retinal vascular lumen and the neural retina. The outer BRB, on the other hand, is formed by retinal pigment epithelium (RPE) TJs to limit the exchange of nutrients and fluid between the choroid and the outer retina. Together, these components play a critical role in preserving retinal homeostasis and immune privilege for optimal visual function [4,5].

Disruption of BRB and vascular leakage disturbs retinal homeostasis with exudates and vasogenic fluid retention, causing retinal neural tissue damage in vascular eye disease. Many factors (hypoxia, oxidative stress, hyperglycemia, and inflammation) can contribute to the breakdown or degeneration of BRB under pathological conditions, eventuating in vision loss. In 2020, global blindness was highly attributable in part to retinal vascular diseases, namely age-related macular degeneration (AMD), diabetic retinopathy, and glaucoma [6]. Breakdown of iBRB has been implicated in diabetic retinopathy [7], retinopathy of prematurity [8], uveitis [9], retinal vein occlusion [10], retinoblastoma [11], and Coats’ disease [12], while oBRB impairment has been linked with neovascular or wet AMD pathogenesis [13]. Two rare congenital vascular disorders, familial exudative vitreoretinopathy (FEVR) and Norrie disease, both associated with mutations in Wnt/β-catenin signaling axis, also manifest iBRB impairment [14,15,16]. Moreover, retinal capillary deficit and BRB impairment have also been observed in human neurodegenerative diseases [17].

In the past few decades, research advances on iBRB have benefitted from development of various animal models mimicking human vascular diseases with iBRB breakdown in rare eye diseases and diabetic retinopathy. These models allowed the discovery of many, if not all, underlying signaling pathways involved in compromised barrier integrity. With these preclinical animal models, various in vivo and ex vivo methodologies with fluorescent or colorimetric molecular tracers are commonly used to directly image and quantify BRB integrity and the extent of extravasation in the retinas. Improvements in image analysis algorithms were also made recently, allowing more accurate quantification of BRB leakage. In addition, many in vitro cell culture models are utilized to allow functional assessment of endothelial barrier properties in vascular endothelial cells or in combination with other cell types. Together these in vivo and in vitro models have also been valuable in evaluating the efficacy of potential therapeutic drugs to restore barrier integrity. This review highlights the classical animal models and methods frequently used to assess iBRB impairment, in addition to summarizing recent advances in this field. We also provide brief discussion of the respective benefits and potential limitations of each model and method, with the goal of helping researchers choose their desired models and methods to suit the nature of their investigation.

## 2. Overview of iBRB: Development, Composition, and Signaling Pathways

### 2.1. Development of iBRB

In the embryonic stage, the developing lens and retina are nourished initially by a transient hyaloid vascular bed, which later regresses alongside retinal vasculature development. Hyaloid regression occurs during the mid-gestation period in humans and postnatally in mice, concomitant with retinal vascular bed formation [18,19].

In humans, retinal vessel sprouting occurs at gestational week (GW) 18, proceeded by retinal vasculogenesis from GW 14–18 from pre-differentiated vascular patent cells around the optic nerve head, with complete vascularization of the peripheral retina just before birth. In mice retinal angiogenesis begins after birth, with complete formation of the superficial plexus by postnatal day (P) 7–8 [20]. The superficial capillaries forms vertical endothelial tip cells and vascular sprouts which dive towards the outer retina and enter the outer plexiform layer to form the deep plexus during P7–12, followed by additional sprouting to form the intermediate capillary network by P14–21 [21,22], to complete a retinal vasculature with three distinct layers.

Soon after the initiation of retinal angiogenesis, development of iBRB occurs involving the formation of TJs between vascular endothelial cells and the suppression of endothelium transcytosis. Current knowledge of this process comes largely from studies in mouse retinas. Endothelial TJs are formed once the adjacent endothelial cells (ECs) come into contact with each other, hence functional tight junctions start to develop as early as P1 in mice following superficial retinal vessel formation. Yet these newly formed vessels still exhibit unrestricted transcytosis marked by the presence of transcytotic vesicles [23]. The vascular–endothelial barrier matures gradually in a proximal-to-distal manner, with fully functional barrier formation in both tight junction formation and limited transcytosis in the mature vessels of superficial layer by P10, yet the nascent sprouting retinal vessel tips still lack an intrinsic functional barrier [23]. Immature BRB permeability until P9 is entirely due to the unrestricted bulk transcytosis with presence of fully functional TJs [23]. Moreover, endothelium transcytosis is gradually suppressed by proximal-to-distal expression of MFSD2A, a transcytosis regulatory protein. MFSD2A is expressed in central mature vessels but not distal leaky neovessels at P7, and throughout the superficial vascular bed by P10, suggesting formation of a fully functional iBRB by P10 [23]. By P18 the vertical sprouting vessels also acquire the barrier characteristics [23] and exhibit complete maturation of the paracellular component, coordinated by a canonical Wnt pathway inhibitor, Apcdd1 [24]. These studies highlight the importance of tight junctions and regulation of transcytosis in iBRB formation. Interestingly, in zebrafish, endothelial tip cells in the nascent vascular sprouts are attached with stalk cells through a ring-shaped junction and these cells maintain the barrier integrity through cell–cell junctions while undergoing proliferation during vascular sprouting [25], suggesting a different species-dependent regulatory mechanism of BRB formation.

In addition to junctional formation in vascular endothelium, close interactions between astroglial and ECs are also required for proper iBRB development. Astrocytic progenitor cells emerge from the optic nerve head at P0 in mice and GW14 in humans [26,27]. On one hand, the developing astrocyte network secretes vascular endothelial growth factor (VEGF) and serves as template for the radial migration of retinal vascular ECs towards the peripheral retina for formation of superficial plexus of retinal vessels [27,28,29], with astrocytic end feet forming connections with the abluminal surface of blood vascular endothelium as part of the iBRB. On the other hand, retinal astrocyte maturation is also dependent upon retinal angiogenesis in a feedback fashion [30,31], where disrupted angiogenesis results in abnormal astrocyte migration and proliferation [32]. Another type of perivascular cells, pericytes, ensheath the retinal vessels even before the BRB is functional, and their depletion is associated with BRB breakdown, abnormal vascular sprouting and re-vascularization, implicated in several retinal diseases such as diabetic retinopathy [22,33,34]. Hence recruitment of neurovascular unit cells is critical for retinal vascularization and formation of functional iBRB.

### 2.2. iBRB Composition

ECs line the inner surface of blood vessels in the retina and CNS as a monolayer encircling the lumen with three unique features and without fenestration, thereby differing from fenestrated choroid and peripheral vascular beds elsewhere that allow free passage of substances [35]. First, adjacent retinal ECs are connected by highly complex tight junctions within intercellular cleft, which regulate the paracellular flux and maintain cell polarity. Second, normal retinal ECs exhibit a limited rate of transcellular vesicular transport (transcytosis) with few transcytotic vesicles, thereby regulating the selective transcellular transport across them [36,37]. Third, the retinal ECs express low levels of vesicle transporters and high levels of efflux pumps, which together further regulate transcellular transport across the blood capillaries [21]. These features of retinal ECs, along with the surrounding pericytes, astrocytes, Müller cells, and microglia, collectively termed as the neurovascular unit, confer the characteristic barrier properties to the retinal and CNS vessels. The pericytes surrounding ECs are ensheathed by astrocytic processes, forming a continuous layer with basal lamina at the abluminal surface of endothelium. Together these endothelial and peri-endothelial characteristics selectively regulate molecular transport across capillaries via two different routes: the paracellular pathway, which is highly regulated by the inter-endothelial junctions; and the transcellular pathway including passive diffusion and ATP-dependent vesicular or non-vesicular transport. Additionally, the neurovascular components stabilize vessels and coordinate blood flow in response to retinal metabolic demands [21,36,38].

### 2.3. Molecular Markers of iBRB and Routes of Transport

#### 2.3.1. Paracellular Transport Is Regulated by Junctions between ECs

Adjacent ECs of the retina and CNS are sealed together by specific junctional protein complexes including tight junctions, adherens junctions (AJs), and gap junctions. These junctions regulate the passage of water, small lipophilic molecules, and blood cells across the intercellular cleft in paracellular transport. Among them TJs and AJs primarily contribute to barriergenesis [21].

##### Characteristics and Components of Tight Junctions

Unlike epithelial TJs which are apically concentrated and have desmosomes, membrane anchors for cytoskeleton filaments, TJs in endothelial cells are often entangled with adherens and gap junctions throughout the intercellular cleft, allowing strict regulation of flux. In addition, most ECs lack desmosomes, although desmosome-like structures are still present in ECs of veins and the lymphatic system [39,40]. Compared to other tissues, CNS and retinal ECs exhibit maximum TJ strands with the smallest paracellular space [36]. TJs, therefore, function as paracellular gates restricting diffusion, thus preventing lateral diffusion of lipids and proteins, and conferring cell polarity. They are also involved in intracellular signal transduction pathways impacting cell proliferation and differentiation [36,37,41].

Under an electron microscope, the TJ complex structure appears like a fibrillary meshwork with an electron-dense cytosolic junctional plaque. The meshwork is formed by transmembrane proteins, including the tetra-span claudin family and MARVEL domain proteins (MAL and related proteins for vesicle trafficking and membrane link), including occludin, the tri-span blood vessel epicardial substance (BVES) protein, and the single-span junctional adhesion molecules (JAM) family proteins. The junctional plaque is composed of adaptor proteins, such as zonula occludens (ZO-1, ZO-2, ZO-3), MAGI proteins, multi-PDZ (MPDZ) proteins, PALS1-associated tight junction (PAT), and cingulin [21,42].

##### Claudin-5 Is One of Major Molecular Markers of BRB and BBB

The claudin protein family constitutes the major TJ structural component and is primarily involved in regulating paracellular transport [43]. Encompassing 27 members [44], claudins can be further subdivided based upon function as pore-forming (claudin-2, -10, -16, -17, -19) and sealing claudins (claudin-1, -3, -5) [45,46]. Expression of claudin-1, -3, -5, and -12 has been reported in retinal and brain endothelium, with claudin-5 (Cln5) being the most abundantly expressed [8,47,48].

In certain vascular pathologies and diseases, altered distribution, expression, or structure of TJ components has been observed. Studies focusing on BRB or BBB loss in ischemia [49,50] and CNS inflammation [51] have demonstrated increased vascular permeability due to Cln5 disruption, indicating its role in maintaining CNS barrier integrity. Cln5 knockout mice exhibit altered size-selective BBB permeability and die shortly after birth [52]. Moreover, dysregulation of Cln5 is implicated in several neurological disorders, such as, chronic traumatic encephalopathy, psychiatric disorders including schizophrenia [53,54], along with suppression of occludin, ZO-1, VE cadherin resulting in increased vessel leakage in several retinal vascular diseases [8,49,55], as well as in impaired BRB in FEVR and Norrie disease models [56,57]. Additionally, persistent suppression of Cln5 in iBRB in the inner retina induces retinal pigment epithelium (RPE) atrophy in the outer retina, when its cycling regulation by a circadian clock gene BMAL1 is disrupted causing metabolic mismatch that overwhelms RPE function [58]. Besides claudin, other transmembrane proteins such as occludins, ZO-1, and JAMs also play important roles in barriergenesis. Reduced retinal occludin expression resulting in increased iBRB permeability has been reported both in rodent model of diabetic retinopathy [59] and in vitro studies [60], while JAM-C maintains TJ integrity [61]. ZO-1 functions as a scaffold protein and regulates cell migration, angiogenesis, and intrinsic tension at endothelial cell junctions [62] and its importance in barrier endothelium organization was demonstrated by the fact that ZO-1 and ZO-2 null mice are embryonically lethal [63].

#### 2.3.2. Transcellular Transport across Endothelium Relies Primarily on Caveolar Transcytosis

Movement of large lipophilic, hydrophilic molecules and ions across the retinal/brain ECs relies on energy-dependent transcellular transport including receptor- and carrier-mediated transporters, pinocytotic vesicles, ion transporters, and pumps [21]. To restrict molecular exchange, ECs lining the retinal and brain capillaries express more efflux pumps and fewer transcytotic mediators, hence maintaining a low rate of transcytosis [64]. Functional BRB/BBB development relies upon suppression of EC transcytosis, hence low levels of caveolae and albumin receptors in barrier endothelium further strengthens the barrier properties [23,65].

Caveolin-1: EC transcytosis primarily occurs through lipid-enriched plasma membrane invaginations of caveolae (or plasmalemmal vesicles). Caveolins (Cav-1, -2, -3) and cavins (cavin-1, -2, -3, -4) form the protein coat of caveolae and among them cav-1 is expressed in cardiovascular endothelium caveolae. Cav-1 and -3 are both required for caveolae formation [66]. In the neural retina, Cav-1 is expressed in endothelial and mural vascular cells, Müller glial cells, and photoreceptors [67]. Cav-1 maintains BRB integrity as ablation of Cav-1 resulted in significant retinal vascular hyperpermeability, especially in branch veins, without affecting junctional proteins [67]. *Cav-1^−/^^−^* mice show resistance to drug-induced albumin leakage and edema in pulmonary vessels [68]. Moreover, Cav-1 expression is induced by VEGF and is upregulated in retinas from a mouse model of STZ-induced diabetic retinopathy [48]. Increased number of caveolar vesicles was also observed on pericyte abluminal surface in diabetic rat retina, suggesting dysregulated Cav-1 in pathological BRB breakdown. In addition, Cav-1 dysregulation has also been linked with cerebral ischemic injury in mice [67]. 

PLVAP: Plasmalemma vesicle-associated protein (PLVAP) is an EC-specific caveolae structural protein. Its upregulation and co-localization with barrier-defunct capillary sites have been associated with increased rates of caveolae-mediated transcytosis in diabetic retinopathy (DR), VEGF-induced retinopathy, brain tumors [69,70]. In intact BRB, expression of PLVAP in retinal ECs is low, consistent with the few observed transcytotic vesicles, yet its expression increases with pathological BRB loss such as in DR, Norrie disease [71,72]. PLVAP influences VEGF-dependent EC vascular permeability, as its expression is induced by VEGF in vitro and in vivo and suppression of PLVAP prevents BRB breakdown by inhibiting VEGF-induced caveolae formation [73]. Additionally, PLVAP expression in leaky microvessels with disrupted barriers is inversely related with TJ protein expression, suggestive of potential interaction between both the transport pathways [16,56,57,67,74].

MFSD2A: Another BRB marker is major facilitator superfamily domain-containing 2a (MFSD2A), a transmembrane protein involved in transportation of docosahexaenoic acid (DHA), a form of omega-3 polyunsaturated fatty acid, in the brain and retinal photoreceptors, which is important for their development and function [75,76]. Intact vascular endothelium in CNS and retinal RPE exhibit high expression of *Mfsd2a* as compared to peripheral ECs, corresponding to the comparative lower rate of transcytosis, crucial in BRB/BBB formation and maintenance [65,75,76,77]. MFSD2A-mediated DHA transport results in increased levels of DHA-containing phospholipids that displaces plasma membrane Cav-1, thereby modifying lipid raft composition, disrupting caveolae formation and hence suppressing caveolae-based transcytosis [77,78]. Increased BBB permeability has been observed in *Mfsd2^−/^^−^* mice exhibit increased BBB permeability [77], yet controversy exists as few other studies have shown its deficiency from retinal vascular endothelium does not seem to affect BRB permeability and photoreceptor function [75,76]. Whether this discrepancy reflects different mouse background or assay methodologies will await further investigation.

### 2.4. Signaling Pathways in BRB Impairment

#### 2.4.1. Norrin/Fzd4/Lrp5/β-Catenin Signaling in Vascular Endothelium

The significance of Wnt/β-catenin signaling in eye development, retinal angiogenesis, and barriergenesis in retinal and brain endothelium has been well documented in many studies. Wnt pathway is activated upon binding of Wnt ligands to high affinity Frizzled (Fzd) receptors and coreceptors LDL receptor-related protein (Lrp5/6) [79,80]. Activation of canonical Wnt/β-catenin pathway causes stabilization and consequent nuclear translocation of β-catenin, eventuating in target genes transcription after β-catenin binding with lymphoid enhancer factor/T cell factor (Lef/Tcf) family [81].

Wnt signaling deficiency has been associated with defective vascular development and loss of functional BRB/BBB [14,82,83]. In humans, two rare pediatric eye diseases, familial exudative vitreoretinopathy (FEVR) and X-linked Norrie disease, correspond to the loss-of-function mutations in Norrin/Fzd4/LRP5 axis and/or several other genes and norrin (Ndp) gene, respectively. These vascular diseases are characterized by severe retinal hypovascularization and persistence of hyaloid vasculature graduating in blindness [84,85,86]. These diseases are well represented by Ndp/Lrp5/Fzd4/Tspan12 knockout mouse models, exhibiting incomplete and abnormal vasculature and absence of deep vascular layer [56,87,88,89]. Also, EC-specific LRP5 or Fzd4 knockout mice exhibited loss of plasticity in BRB/BBB ECs along with barrier loss and defective angiogenesis [90], demonstrating a vascular endothelium specific role of Fzd4/LRP5 receptor signaling in retinal angiogenesis and barrier formation. On the other hand, norrin protein, a high affinity Fzd4/LRP5/Tspan12 ligand, is produced by retinal Müller glia [91] to mediate the interaction between glial cells and vascular endothelium.

Both LRP5- and Norrin-deficient retinas had remarkably downregulated TJ protein Cln5 in their retinal ECs corresponding to the increased vascular permeability, as well as increased PLVAP expression [16,56,57,83]. Moreover, Lrp5- and Ndp-deficient mice both exhibit low levels of MFSD2A and high levels of Cav-1, leading to high transcytosis levels and increased BRB permeability, which was restored after *Mfsd2a* overexpression [92], indicating that regulation of retinal EC transcytosis was mediated through the Norrin/Fzd4/β-catenin axis by direct upregulation of *Mfsd2a*. These studies demonstrate that Wnt/β-catenin pathway is crucial in BRB maintenance, by regulating both paracellular as well as transcellular pathways (Figure 1).

Recently, inhibition of Wnt signaling enhanced transient opening of BRB and improved MFSD2A-dependent drug delivery in glioma through EC transcytosis [93]. Activation of Wnt signaling also mitigated BRB dysfunction induced amyloid-β in Alzheimer’s disease [94], and brain edema following intracerebral hemorrhage in mice [95]. These studies suggest promising directions of targeting Wnt signaling for therapeutic applications in BRB restoration or manipulation.

#### 2.4.2. Hypoxia-Induced VEGF Signaling: Interaction between Glia and Vascular Endothelium

The importance of VEGF in developmental angiogenesis, retinal vascular pathologies as well as in cancers has been intensively studied in the past several decades and summarized in many reviews [96]. VEGF-A is the dominant member of VEGF family in angiogenic regulation, whereas a few other forms are involved in lymphangiogenesis, such as VEGF-C and -D [96]. VEGF-A secretion is hypoxia-inducible factor 1 (HIF1)-dependent [97,98], and in the retina it is produced by various retinal cells, including two main types of glia cells: Müller glia and astrocytes [99], as well as microglia [96]. VEGF-A signals mainly through two tyrosine kinase receptors VEGFR1 and VEGFR2, both highly expressed in the vascular endothelium and also retinal neurons [100,101], with VEGFR2 being the main receptor for signaling [96]. During retinal development, a hypoxia-induced VEGF gradients guides the formation of the superficial retinal vascular network following an astrocyte template [29]. VEGF is also important for maintenance of normal healthy mature vessels in the retina and choroid [96].

Upregulation of VEGF is the major cause of pathological angiogenesis and vascular permeability in proliferative retinopathies including diabetic retinopathy and retinopathy o prematurity, and in neovascular AMD [102,103] (Figure 1). Both Müller cells and astrocytes contribute to HIF-dependent increase of abnormal VEGF levels under pathological conditions [98]. Increased VEGF induces paracellular permeability via downregulation of occludin and ZO-1, leading to increased paracellular permeability across iBRB [49,59]. Suppression of Cln5 and occludin mediated by VEGF upregulation has also been associated with barrier breakdown in mouse models of diabetic retinopathy and oxygen-induced retinopathy (OIR) modeling retinopathy of prematurity [8,55,73]. Moreover, VEGF also increases caveolae formation and transcytosis by inducing Cav-1 and PLVAP expression [48,73]. Because of its predominant role in regulating angiogenesis and vascular permeability, anti-VEGF therapies have become the standard of care in neovascular AMD, proliferative retinopathies, and diabetic macular edema.

#### 2.4.3. PDGFB, TGF-β and Ang1/Tie2 Signaling: Cross Talk between Pericyte and Vascular Endothelium

Pericytes ensheath the retinal capillaries and support their development and repair. In barrier ECs, the pericytes:EC ratio (1:1 in retina; 1:4 in brain) is much higher than vascular beds in other organs and are required for barrier formation [104,105]. Pericyte cell dysfunction or death was found in aging, several neurovascular and degenerative diseases, such as DR, and Alzheimer’s disease [104]. Mice lacking pericytes (PDGF-B and PDGFR-β null mice) are embryonically lethal and demonstrate defective TJs, increased transcytosis, leaky vessels, and neurodegeneration [34,105,106].

During development, nascent retinal vascular endothelial tip cells recruit pericytes through platelet-derived growth factor-B (PDGF-B)/PDGF receptor-β (PDGFR-β) signaling, essential for angiogenesis and BRB maintenance [107,108]. Failure to recruit pericytes in the developing mouse retina leads to BRB disruption; however, loss of pericytes in stable adult mouse retinal vessels does not affect BRB integrity, although the retinal vessel sensitivity to VEGF-A was shown to be increased, causing angiopoietin-2 (Ang2) upregulation and DR-like vascular pathology [34] (Figure 1). This suggests that PDGF-B/PDGFR-β signaling is critical for BRB formation and maturation and may contribute to retinopathy pathology, yet it is not indispensable in adult mature vessels [34].

Pericyte recruitment is followed by EC-pericyte adhesion, mediated by transforming growth factor-β (TGF-β) signaling. TGF-β is produced by astrocytes, Müller glia, as well as other perivascular cells [109,110]. Dysregulation of TGF-β is associated with many pathological features, such as loss of retinal barrier, defective angiogenesis, inflammation, and tissue fibrosis [111]. TGF-β signaling induces upregulation of potential angiogenic modulators, such as VEGF, fibroblast growth factor (FGF), PDGF, in neovascularization [111]. TGF-β promotes N-cadherin upregulation via Smad pathway activation, where Smad4-ablated ECs shows pericyte detachment and barrier leakage [112]. TGF-β signaling inhibits retinal EC migration and proliferation, thus sustaining BRB integrity [113].

Pericytes also stabilize the blood vessels and maintain BRB via angiopoietin (Ang)/tyrosine kinase receptor (Tie-2) signaling. Ang-1 secreted by pericytes promotes blood vessel stabilization, maturation, and remodeling as well as barrier differentiation via autophosphorylation of Tie-2, secreted by ECs, whereas Ang-2, another EC-secreted Tie-2 ligand, destabilizes ECs and stimulates VEGF-induced angiogenesis [21,36]. Disrupted levels of Ang-1 and Ang-2 may contribute to vascular pathology mimicking DR [114,115], and hence represent potential targets in designing retinal vascular disease treatment [116,117].

#### 2.4.4. Sonic Hedgehog (Shh) Signaling: Glial Cells

In addition to the roles of retinal glial cells in regulation of VEGF, Norrin, and TGFβ secretion and signaling, another astrocyte-dependent signaling pathway involved in modulating EC barrier is Shh signaling, crucial for various developmental processes, such as axon guidance, neuronal differentiation, vessel development, and angiogenesis [118,119]. Under normal conditions in the absence of Shh, its receptor Patched-1 (Ptch1) represses Shh coreceptor smoothened (Smo). Upon activation, Shh binds Ptch1, leading to the release of Smo inhibition and its subsequent nuclear translocation to further activate target genes including Gli transcription factors. Shh contributes to EC barrier properties by upregulating TJs, occludin and Cln5, as Shh-deficient mice exhibit defective BBB formation with significant loss of TJ proteins and embryonic lethality [104]. Recently, the role of Shh signaling in regulating VEGF in OIR mouse model has also been documented [120]. Also, deficiency of another marker, dystrophin, Dp71, in Müller cells has been associated with BRB breakdown [121]. Furthermore, glial cells mediate neurotrophic responses by synthesizing several neurotrophic factors, some of which are involved in angiogenesis, such as basic FGF(bFGF), and glial-derived neurotrophic factor (GDNF) [122].

## 3. Mouse Models with Inner Retinal Vascular Leakage and iBRB Impairment

Over the years, many rodent models have been developed which exhibit retinal vascular leakage and impaired iBRB. These include models with identical disease genes to humans that recapitulate various human vascular eye diseases, and models with targeted mutation of genes important for maintaining intact retinal vasculature and iBRB, such as VEGF and Wnt signaling. Together, these animal models have greatly enhanced investigation into the basic molecular mechanisms in iBRB regulation and pathogenesis of human eye diseases affected by abnormal retinal angiogenesis and iBRB breakdown.

### 3.1. Mouse Models of Norrie Disease and FEVR Exhibit Defective Ocular Angiogenesis and Impaired iBRB

In humans, two rare congenital disorders of retinal angiogenesis, X-linked Norrie disease, and FEVR, show converging ocular manifestations. Male infants affected by Norrie disease are often blind either at birth or in early infancy and exhibit retinal folds/detachment, pseudoglioma, and vitreous hemorrhage. A smaller percentage of patients may also develop cochlear defects, hearing loss, and mental retardation, in addition to ocular defects [84,89,123]. A similar but milder disease FEVR, first described in 1969 by Criswick and Schepens [85], is characterized with overlapping retinal features including peripheral retinal hypovascularization, absence of deep retinal vessels, retinal folds/detachments, vitreous hemorrhage, and subretinal exudates. Norrie disease is associated with loss-of-function mutations in Norrin (*NDP*) (>100 mutations reported) whereas FEVR is caused by mutations in multiple genes including *NDP*/*FZD4/LRP5/TSPAN12/ZNF408* [124,125,126,127,128] and recently reported *CTNNA1* [129] and *KIF11* [130]. Additionally, *LRP5* mutations may also cause osteoporosis and osteopenia [131]. The mutation can be X-linked (*NDP* mutation), autosomal dominant (*ZNF408* mutation), or autosomal dominant and recessive (*FZD4/LRP5/TSPAN12* mutations). Both diseases develop due to impaired retinal angiogenesis and persistent hyaloid vessels, causing blindness [132,133].

Wnt signaling was discovered as a major pathway underling Norrie disease and FEVR, since the seminal discovery of Norrin protein as a Wnt ligand [134]. Canonical Wnt signaling cascade is activated upon binding of Wnt ligands frizzled (FZD) receptor and LRP5/6 coreceptors with disheveled (Dvl) recruitment which stabilizes β-catenin and allows its translocation into the nucleus, where it triggers transcription of target genes upon binding with Lef/Tcf family of transcription factors [79]. Canonical Wnt signaling is involved in neuronal development [135], blood–brain/blood–retinal barrier formation [136], vascular EC apoptosis and hyaloid regression [137], and retinal vessel angiogenesis [55]. Apart from Wnt ligands, Norrin also exhibits structural similarity to Wnt ligands and binds specifically with FZD4/LRP5 activating the Wnt/β-catenin axis [134,138].

Canonical Wnt signaling is disrupted when β-catenin is abolished, or its stabilization is prevented. Mouse models with genetic deficiency of Norrin or LRP5 exhibit disrupted Wnt signaling resulting in similar retinal vascular abnormalities as seen in human Norrie disease and FEVR [56,90,134]. Mice lacking LRP5 (*Lrp5^−/^^−^*) show delayed development of the superficial capillary plexus and persistent hyaloid vessels, and absence of intermediate and deep retinal capillary beds (Figure 2A). The superficial retinal capillaries of these mice are dilated with microaneurysm-like vascular lesions starting from P12 through adulthood, corresponding to EC clustering and stalled deep layer migration, and exhibit reduced vascular density in the brain [55,56,90]. Activation of canonical Wnt pathway in *LRP5^−/^^−^* mice partially rescues the developmental retinal vascular defects [139]. Loss-of-function mutations in Norrin (*Ndp^y/^^−^*) or Fzd4 (*Fzd4^−/^^−^*) mouse model also leads to similar ocular vascular defects, such as lack of two intraretinal capillary beds with enlarged, tortuous superficial vessels, failed regression of hyaloid vasculature, fenestrations in several vitreal face vessels, and retinal hemorrhage [134,140]. These models of impaired Norrin/Fzd4/Lrp5 signaling demonstrate manifestation of hypoxia owing to the lack of inner retina vasculature resulting in upregulation of HIF-1α, VEGFA as a compensatory response, leading to formation of glomeruloid structures in the peripheral plexus [134,140,141]. In addition to defective angiogenesis, these models with deficient Wnt signaling also demonstrate compromised BRB and increased vascular leakage (Figure 3A), modulated in part by suppression of Cln5 andMfsd2A transporter, increased PLVAP expression [16,55,56,83,90]. In addition, increased vesicular transcytosis and retinal vessel fenestrations can be observed under electron microscopy in *Ndp^y/^^−^* eyes [92] (Figure 4). These defective vascular features have shown to be restored to normal vasculature by stabilization of β catenin [16,142,143].

Defective retinal angiogenesis impacts visual function in Wnt-deficient mice. Both *Lrp5^−/^^−^* and *Ndp^y/^^−^* mice had reduced ERG *b*-wave [139,144] and full-field ERG *b*-wave, respectively [145], suggestive of hypoxic microenvironment development and functional impairment of inner retina. *Ndp^y/^^−^* retinas showed additional complete absence of oscillatory potential, indicating retina ganglion cell functional deficit [145]. TSPAN12, a transmembrane protein binding Norrin and FZD4/LRP5, modulates retinal vessel development by activating Norrin/β-catenin axis [87,146] and mutations in *TSPAN12* have been associated with FEVR in humans [147]. TSPAN12 knockout mice shows similar phenotypes, such as persistent hyaloid, absence of deep vascular bed, vascular malformations in superficial and intermediate retinal plexus [87] and vision function defects with diminished ERG *b*-wave [148]. Also, EC-specific inactivation of FZD4, LRP5 or TSPAN12 results in phenotypes similar to systemic mutants, suggesting an EC intrinsic role of Wnt signaling in angiogenesis and iBRB regulation [87,90,146].

**Figure 2 cells-12-02443-f002:**
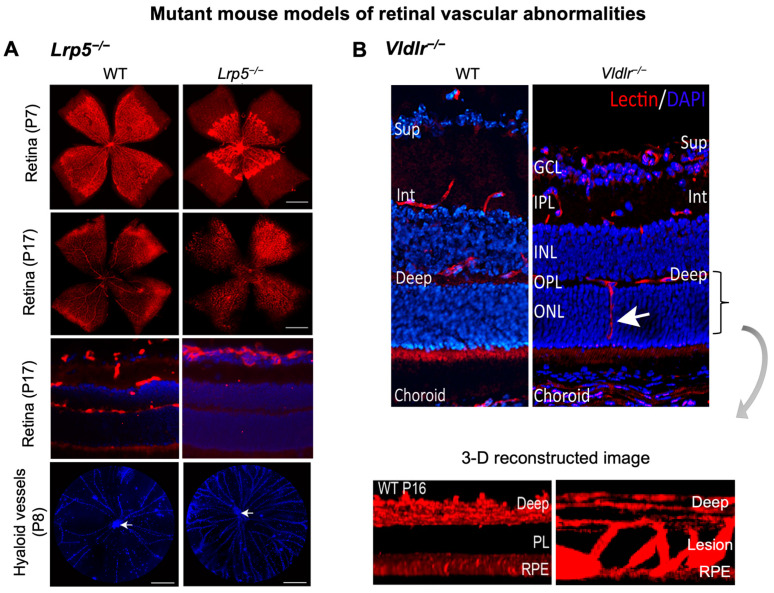
**Retinal vascular abnormalities in two mouse models with impaired iBRB.** (**A**) Compared with wild-type (WT) eyes, LRP5 knock out (*Lrp5^−/^^−^*) mice have deficient Wnt signaling and exhibit incomplete superficial vasculature during development and absence of both intermediate and deep vascular plexus. Retinal flat mounts were stained with blood vessel marker isolectin IB_4_ (red) and whole eye cross-section were stained with isolectin IB_4_ (red) and nuclear stain DAPI (blue) in P17 retinas. *Lrp5^−/^^−^* eyes also have persistent hyaloid vasculature as visualized by higher number of DAPI-stained hyaloid vessels at P8 with respect to WT mice (white arrows: hyaloid artery). (**B**) *Vldlr^−/^^−^* mice are a model of spontaneous intraretinal and subretinal neovascularization with aberrant vertical sprouting of blood vessels (white arrow) from deep retinal vascular bed towards the normally avascular photoreceptors layer into the subretinal space. Lower panel shows the 3D reconstructed image of vascular lesions (magnified view) in *Vldlr^−/^^−^* mice retina as the compared with the absence of lesions seen in WT retinas. Sup, superficial vascular layer; Int, intermediate vascular layer; Deep, deep vascular layer; GCL, ganglion cell layer; IPL, inner plexiform layer; INL, inner nuclear layer; OPL, outer plexiform layer; ONL, PL: photoreceptor layer; RPE, retinal pigmented epithelium. Images were adapted with permission from [20,149].

**Figure 3 cells-12-02443-f003:**
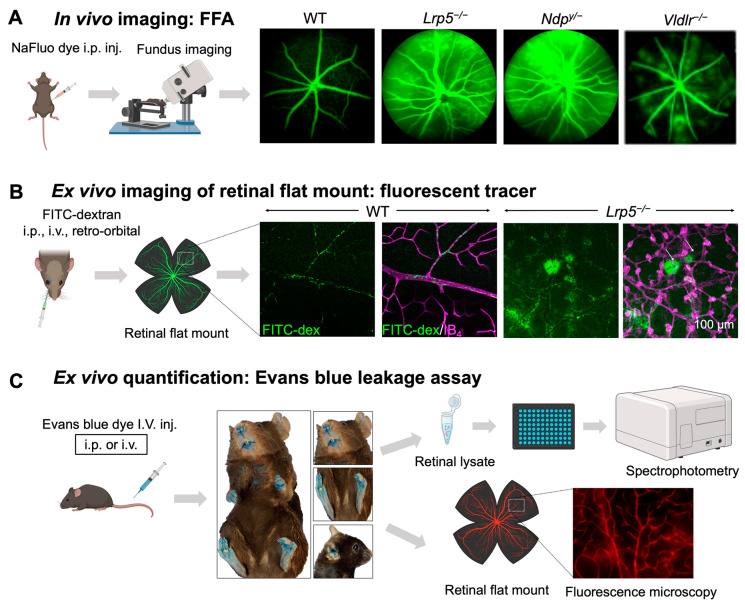
**In vivo and ex vivo methods for assessing iBRB integrity in mouse models.** (**A**) Schematics of in vivo imaging procedure with FFA and representative images. Mice were intraperitoneally injected with sodium fluorescein (NaFluo) dye (green) and fluorescein-filled blood vessels with associated leakage were immediately visualized using fundus camera. Sequential fundus images were taken at different time points from 0–10 min, to capture the process of retinal vascular leakage. WT mouse fundus showed intact blood vessels while *Lrp5^−/^^−^*, *Ndp^y/^^−^*, and *Vldlr^−/^^−^* fundus exhibited leaky retinal vessels due to their impaired iBRB. (**B**,**C**) Ex vivo methods of vascular leakage imaging and assays with FITC-dextran or Evans blue staining. (**B**) FITC-conjugated dextran of varying molecular weight (here 70 kDa) was injected via retroorbital plexus into the mice. The extravasated FITC-dextran dye (green; white arrows) in *Lrp5^−/^^−^* eyes was visualized in retinal flat mounts co-stained with blood vessel marker isolectin IB_4_ (magenta) with glomeruloid vascular structures, while WT mouse retinas showed FITC-dextran confined within normally branched blood vessels. The leakage intensity of extravasated FITC-dextran complex can be quantified as a measurement of vascular leakage. (**C**) Extravasation of Evans blue dye is a popular index of iBRB disruption or altered vascular permeability. Evans blue was intravenously administered in mice, which resulted in bluish coloration of tissues, clearly seen in less pigmented body areas like paws, ears, and the snout region. The extravasated dye can be either quantified via absorbance or fluorescent spectrophotometer or visualized in retinal flat mounts under a fluorescent microscope. FFA, fundus fluorescein angiography; NaFluo, sodium fluorescein; inj, injection; FITC-dex, fluorescein isothiocyanate conjugated dextran; IB_4_, isolectin B_4_. Scale bar: (**B**) 100 μm. Images in panels (**A**,**B**) were adapted with permission from [92,149]. Parts of the figure were created with BioRender.com.

**Figure 4 cells-12-02443-f004:**
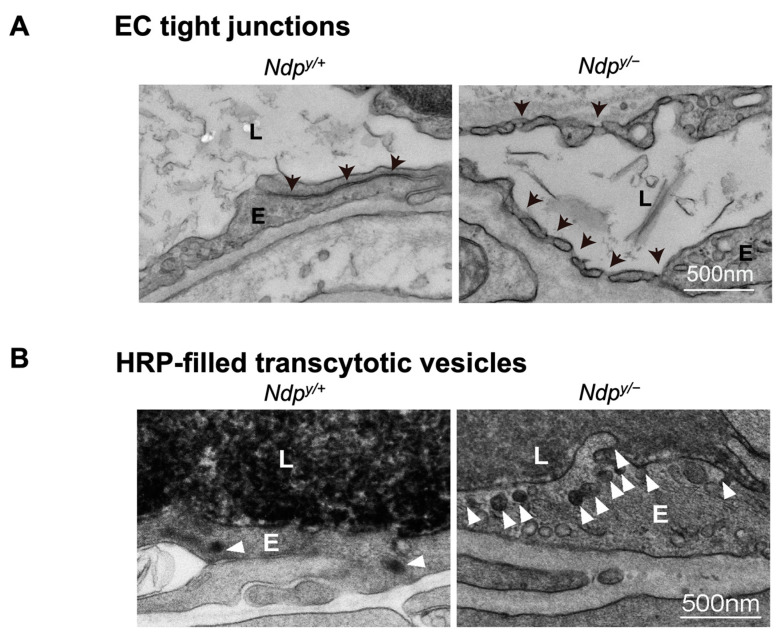
Ultrastructural changes associated with impaired iBRB and HRP-based ex vivo detection of transcytosis under electron microscopy. (**A**) Intact tight junctions (black arrows) between adjacent retinal vascular endothelial cells in wild-type (*Ndp ^y/+^*) mouse retinas were imaged with transmission electron microscopy (TEM), whereas absence of tight junctions and presence of fenestrations (black arrowheads) were seen between adjoining endothelial cells in norrin deficient (*Ndp^y/^^−^*) mice. (**B**) TEM images showing HRP (an enzymatic leakage marker)-filled transcytotic vesicles within vascular ECs. Ultrathin cross sections from HRP injected (retro-orbital route) *Ndp^y/+^* and *Ndp^y/^^−^* eyes were imaged under TEM to visualize transcytotic vesicles filled with HRP. Under normal conditions, EC transcytosis across the CNS barriers (iBRB/BBB) is highly suppressed. *Ndp^y/+^* retinal vascular endothelium shows few HRP-filled vesicles (white arrowheads) whereas *Ndp^y/^^−^* retinas exhibit more transcytotic vesicles, suggestive of increased transcytosis and leaky iBRB. EC, endothelial cells; TEM, transmission electron microscopy; HRP, horseradish peroxidase. Scale bar: (**A**,**B**) 500 nm. Images were adapted with permission from [92].

Recently, additional new genes, like *CTNNA1* mutation have been associated with FEVR, that acts by disrupting cadherin–catenin complex and thereby, over activating Norrin/β catenin signaling [129]. Integrin-linked kinase control (*ILK-1*), another recently discovered FEVR diseases gene [150], is involved in cell–matrix interactions and retinal angiogenesis. EC-specific *ILK* knockout mice are embryonic lethal due to impaired vascularization, highlighting their crucial role in retinal angiogenesis [150]. Inducible EC-specific Ilk knockout (*iIlk^iECKO^*) mice shows FEVR-like vascular defects with compromised retinal sprouting, reduced EC proliferation, defective vascular remodeling, and dysfunctional BRB [150]. Mutations in other genes, such as zinc-finger family DNA binding protein (*ZNF408*) [128] have also been associated with FEVR in humans. Loss of function mutation of *ZNF408* in zebrafish resulted in impaired hyaloid vasculature development, vascular insufficiency and profuse vascular leakage at 180 days post fertilization, indicating its role in retinal barrier development [151]. In addition, null and conditional floxed loss-/gain-of-function mutations in β-catenin have provided significant insights in the role of Wnt signaling in development and cell adhesion. Mice with null β-catenin mutation exhibit gastrulation defects and are embryonically lethal; however, embryos lacking β-catenin revealed intact adherent junctions due to the involvement of γ-catenin in cell adhesion formation [152].

Together these mice with loss-of-function mutations of Wnt/β-catenin axis components represent ideal animal models for exploring pathways associated with pathological retinal angiogenesis and compromised iBRB integrity as seen in Norrie disease and FEVR patients. So far, several studies have reported normalization of Wnt/β-catenin-deficient retinal angiogenic defects in *Ndp^y/^^−^* mice by overexpression of Norrin or β-catenin [16,153], and in *Lrp5^−/^^−^* retina by pharmacological Wnt signaling activation via lithium treatment [142]. Recent studies using FZD4-surrogate treatment to restore BRB/BBB in pathological conditions have also been reported. Treatment with these FZD4-surrogate tetravalent antibodies reversed pathological BBB and BRB leakage, and angiogenesis defects in *Ndp ^y/^^−^* mice [154], while also restoring BRB function and retinal angiogenesis in *Tspan12* deficient mice [155].

### 3.2. Mouse Models of Paracellular iBRB Leakage with Impaired Junction Proteins

The presence of well-developed TJs between adjacent endothelial and epithelial cells is indispensable for the establishment of iBRB and oBRB, respectively. Retinal vascular ECs have specialized TJs that regulate the molecular flux via the paracellular transport, which are composed of multiple junction proteins including Cln5, ZO-1, and occludin. Dysregulation in these TJ proteins contribute to increased vascular leakage with altered paracellular transport.

Several studies have emphasized on the critical role of Cln5 for establishing BRB/BBB permeability in diabetes and ischemic-reperfusion injury [51]. Constitutive Cln5 knockout mice (*Cln5^−/^^−^*) have structurally intact TJs in brain ECs, indicating that TJ strands may consist of other claudin species. However, these KO mice died within 10 h of birth and demonstrated size-selective loosening of junctions with excessive leakage and extravasation of small molecular weight (<800 Da) tracers from barrier ECs [52]. Additionally, isolated brain ECs from Cln5 and occludin double knockdown mice demonstrates leakage of tracer molecules ranging 3–10 kDa [156]. Severe vascular permeability was also observed upon si-RNA-directed targeted Cln5 ablation or inducible EC-specific suppression [157,158]. Another study using TJ-deficient mouse strain with ZO-1 knockout, also shows embryonic lethality around embryonic day (E) 10.5 due to impaired angiogenesis and cell apoptosis [63]. Ablation of ZO-2, another tight junction scaffold protein, in mice also results in embryonic lethality, due to increased apoptosis in early gastrulation (at E7.5), soon after implantation, suggesting requirement of both ZO-1 and ZO-2 during embryonic development. These embryos exhibited altered apical junctional architecture, increased paracellular permeability, and hence leaky TJs [159]. Additionally, the EC-specific role of occludin in regulating retinal vascular permeability by preserving TJ organization has been shown in STZ-induced diabetic mice [49]. Together, these studies indicate the importance of TJ proteins, specifically Cln5 and ZO-1 in embryonic CNS angiogenesis and barrier permeability.

Upregulation of VEGF, a well-known inducer of angiogenesis and permeability, contributes to vascular hyperpermeability and BRB loss in various retinal vascular diseases. Increased VEGF causes downregulation of occludin and ZO-1, leading to increased paracellular permeability across iBRB [59]. VEGF induces phosphorylation, ubiquitination, and further suppression of occludin via activation of protein kinase C (PKC) cascade, and VE-cadherin, thereby affecting iBRB permeability [49]. Suppression of Cln5 and occludin mediated by VEGF upregulation has also been associated with BBB breakdown in a CNS inflammatory mouse model, where recombinant Cln5 showed protective effects [51], and in OIR mouse models [8,55].

### 3.3. Mouse Models of Spontaneous Retinal/Choroidal Neovascularization with Vascular Leakage

In retinal vascular diseases, development of pathological retinal vessel may originate either from choroidal or retinal vascular beds. Wet/exudative AMD is one of the most common causes of vision loss in the elderly and is often characterized by Bruch’s membrane disruption, atypical intravasation of vessels from the fenestrated choroid into the sub-retinal epithelium or subretinal space, and hence macular neovascularization. These newly developed neovessels from the choroid are generally leaky, resulting in sub-RPE or sub-retinal fluid or blood accumulation, RPE detachment and atrophy [160]. Wet AMD is distinguished into three types, where type 1 involves sub-RPE invasion of choroidal vessels, type 2 corresponds to expansion of choroidal vessels into the sub-retinal space following disruption of Bruch’s membrane and RPE, and type 3 is characterized by abnormal proliferation of retinal capillaries into the subretinal space, also known as retinal angiomatous proliferation (RAP), as a variant of wet AMD. RAP is often marked by the presence of retinal-choroidal neovascular anastomosis leading to photoreceptor dysfunction, RPE detachment and vision loss [161,162].

Upregulation of VEGF [163], and deletion of very low-density lipoprotein receptor (VLDLR) [164] and adhesion receptor ADGRF5 [165] have been documented in abnormal ocular angiogenesis. Several animal models exhibit spontaneous ocular angiogenesis with abnormal intraretinal, subretinal, and choroidal neovascularization, imitating the clinical features of these neovascular diseases. They have played a vital role in exploring the associated pathophysiology.

#### 3.3.1. *Vldlr^−/^^−^* Mice

In the eye, VLDLR is expressed by retinal vascular ECs, vascular smooth muscle cells and RPE, where it is involved in negative regulation of angiogenesis, and VLDLR-deficient environment promotes retinal angiogenesis in vivo and vascular endothelial growth in vitro [20]. Alterations in VLDL, VEGF, and LRP6 genes have been associated with AMD in humans [166]. *Vldlr^−/^^−^* mouse was developed in 1995 [167] for initial investigation in lipid regulation, and their ocular vascular abnormalities were characterized as a model for RAP in 2003 [168]. These mutant mice serve as a reliable animal model for studying subretinal neovascularization, retinal angiogenesis and choroidal anastomosis and recapitulate primary features of human RAP and macular telangiectasia (MacTel).

*Vldlr^−/^^−^* mice manifest abnormal vessel sprouting in the outer plexiform layer around P12–14 which then migrate towards the photoreceptors and into the RPE by P16 (Figure 3B). These mice show increased vascular leakage in new vessels [149] and form choroidal anastomosis by 15 days to 3 months of age, leading to subretinal hemorrhages and RPE disruption (by 6 weeks), with increased inflammation, and age-dependent photoreceptor degeneration (by 10–12 months) [168]. *Vldlr^−/^^−^* mice exhibit increased vascular permeability, visualized as several hyperfluorescent spots in fundus fluorescein angiography (FFA) throughout the retina [149] (Figure 4A) and increased fenestrations, particularly in ECs involved in neovascularization. Also, they exhibit increased retina thickness, suggestive of retinal edema [149]. *Vldlr^−/^^−^* retinas show upregulated protein levels of VEGF, bFGF, inflammatory cytokines, with Müller glial cell activation at neovascular lesions [169]. In addition, Wnt signaling dysregulation [164,170], hypoxia and inflammation [171] may also potentiate the *Vldlr^−/^^−^* ocular pathologies. An LRP5/VLDLR double KO mouse model showed absence of subretinal vessel growth, lack of EC migration into deep retinal layers and significant reduction in Slc38a5 transporter, resembling *Lrp5^−/^^−^* phenotype, suggesting that lack of deep layer vessels in the absence of LRP5 signaling prevents formation of subretinal neovascularization in *Vldlr^−/^^−^* mice [164]. Other studies have documented the involvement of dysregulated retinal energy metabolism [172], inflammation [149], protective effects of antioxidants, neurotropic factors [173], resveratrol [169], and nanoceria [174] in suppressing pathological neovascularization in *Vldlr^−/^^−^* mouse. Beyond the retinas, VLDLR along with apolipoprotein E4 (ApoE4) also promotes β-amyloid peptide (Aβ) accumulation and BBB breakdown in mouse brain [175].

#### 3.3.2. VEGF-Overexpressing Transgenic Mice

In 1997, a photoreceptor-specific human VEGF-overexpressing mouse model (VEGF_164_/Rho) was established by incorporating human VEGF cDNA under the control of bovine rhodopsin (Rho) promoter [176]. These VEGF excessive gain-of-function mice demonstrated increased VEGF transgene expression from P6 onwards with neovessels originating from deep retinal vascular layer at P14 extending into the subretinal space by P18 as blood vessel clumps encompassed by proliferated RPE cells, followed by gradual photoreceptor degeneration and inner nuclear layer disorganization. Albeit with intact Bruch’s membrane, multiple areas of retinal vascular leakage corresponding to discrete spots of neovascularization were observed [176,177]. This model clearly demonstrated that VEGF overexpression in photoreceptors is sufficient to induce intra- and sub-retinal neovascular changes [176,178]. In addition, in a separate model using a human Vegf_165_ gene driven by a truncated mouse Rho promoter, low expression of additional human VEGF was sufficient to cause gradual progression of retinal neovascularization [179].

A different VEGF-overexpressing mouse model (*VEGF_164_/RPE65*) was developed using a construct containing murine *VEGF_164_* cDNA driven by murine RPE65 promoter in albino mice [180]. These mice showed increased VEGF protein production by RPE, leading to significantly increased choroidal leakage in intrachoroidal neovascularization yet without penetration through intact Bruch’s membrane, and hence no CNV [180]. Co-overexpression of VEGF and Ang2 in retina or RPE alone does not cause CNV formation, unless used in conjunction with subretinal delivery of adenoviral vector-Ang2 construct [181].

Overall, these and several other studies revealed that VEGF overexpression in the retina is sufficient to induce intraretinal or intrachoroidal neovascularization but not CNV, hence, insults to Bruch’s membrane and/or RPE are essential for CNV development, as observed in AMD [181]. These VEGF-overexpressing mouse models have contributed significantly to studies on molecular changes related to excess retinal VEGF and development of anti-angiogenic therapies. Nowadays, laser-induced CNV models, involving laser-targeted disruption of Bruch’s membrane followed by neovessel infiltration into the subretinal space, are majorly used for modeling spontaneous CNV development [20,182], particularly for investigating other factors independent of VEGF, or exploring VEGF-independent or combinational therapies.

#### 3.3.3. JR5558 or Neoretinal Vascularization 3 (NRV3) Mice

The JR5558 or NRV3 mouse line, showing spontaneous abnormal neovascular lesions and vascular leakage, was discovered by Jackson Laboratory [183] that mimics the early clinical stages of RAP. This strain exhibits spontaneous VEGF-A-dependent neovascularization originating from the deep retinal vascular bed at P15, extending into the sub-retinal space between P17–P25, representing stages I and II of human RAP. These mice develop depigmented fundus lesions and exhibit increased vascular leakage, as detected by FFA and fenestrated neovessels. However, Bruch’s membrane and the choriocapillaris remain intact. Adult mice (3-months old) revealed neovascular lesions in the photoreceptor layer and in subretinal space surrounded by proliferating RPE, retinal edema, photoreceptor cell dysfunction and subsequent cell death, with significantly reduced rod and cone-ERG responses by 8-months age [183,184,185,186]. The vascular abnormalities in this strain were attributed to a single base deletion in *Crb1* or *rd8* allele [186]. These mice have been effectively used for exploring targeted anti-angiogenic therapies for ocular neovascularization pathologies with associated leakage [187,188,189,190].

#### 3.3.4. Figure Eight (*fgt*) Mice or *Ap1g1^fgt^* Mutant Mice

*Fgt* model is a spontaneous retinal vascularization also discovered by Jackson laboratory and linked with the mutation of *Ap1g1* gene that encodes adaptor protein complex AP-1, gamma 1 subunit [191]. Mutations in AP1G1 were linked with neurodevelopmental disorders in humans [192]. Unlike *Ap1g1* null mice which is embryonically lethal, mice homozygous for a hypomorphic mutation of *Ap1g1* exhibit abnormalities in inner ear sensory epithelium, RPE, thyroid follicular epithelium, and germinal epithelium of testis [186,191]. The phenotype develops due to a 6-bp in-frame deletion in exon 13 of *Ap1g1* gene, resulting in deletion of two amino acids in the translated protein. *Ap1g1^fgt^* mutant mice show retinal abnormalities with fundus spotting, and choroidal neovascularization with multiple sites of vascular leakage by 6 weeks; this is followed by scattered areas of depigmentation by 6 weeks that progresses to development of large, depigmented areas by 16 weeks. However, no change in ERG amplitude (until 16 weeks of age) has been reported in these *fgt/fgt* mutant mice [186,191]. These mice may serve as a reliable model to study retinal abnormalities and choroidal vascular anastomosis. Aside from retinal defects, these mice also develop progressive hearing impairment due to gradual loss of inner ear hair cells and organ of Corti degeneration [186,191].

#### 3.3.5. Mouse Models with Age-Dependent Retinal Vascular Neovascularization

Besides the spontaneous/mutant models, several other wet AMD/RAP mouse models exhibiting abnormal age-related retinal neovascularization have also been characterized. Many belly spot and tail (Bst) chromosome 16 mutant mice exhibits age-related subretinal neovessel formation with RPE disruption, retinal detachments, defects in Bruch’s membrane and retinal-choroidal neovessel connections [193]. The incidence of phenotypic expression could be due to the genetic background of the mouse. Almost all ceruloplasmin (Cp) and hephaestin (Heph) double KO (*Cp^−/^^−^*/*Heph^−/y^*) mice displayed age-dependent subretinal neovascularization, RPE hypertrophy, and photoreceptor degeneration [194]. Other AMD-models, such as SOD1*^−/^^−^*, Ccl2*^−/^^−^*/Cx3cr1*^−/^^−^*, ApoE-overexpressing, Ccr2*^−/^^−^*/Ccl2*^−/^^−^* mice exhibited low levels of neovascularization (within 8.3–25%) or less pronounced vascular effects along with drusen formation [195]. Another study using double transgenic mice APP_SWE_/PS1_∆E9_ (ADtg) revealed age-dependent retinal capillary degeneration, PDGFRβ loss with vascular Aβ accumulation, claudin 1 reduction, and substantial increase in iBRB permeability [196]. These models provide additional tools to investigate the effects of aging and pathogenic risk factors of AMD on ocular vascular pathologies and BRB breakdown.

### 3.4. Mouse Models of Diabetic Retinopathy and Diabetic Macular Edema

Formation of blood vessels de novo from dispersed endothelial cells (vasculogenesis) or sprouting of new vessels from existing ones (angiogenesis) are both critical in eye development and maintaining homeostasis, however, abnormal angiogenesis or neovascularization is pathogenic and a hallmark of many vascular ocular diseases [96]. Diabetic retinopathy (DR), a common microvascular complication of diabetes mellitus (DM), is a primary cause of vision impairment in the middle-aged population. BRB impairment and neurovascular unit deterioration are the principle early pathologies for non-proliferative DR. The early non-proliferative stage of DR is characterized by retinal vessel tortuosity, dilation, capillary obstruction, increased vascular leakage [195]. The hyperglycemia-induced retinal vascular dysfunction and vessel loss further leads to retinal ischemia- and hypoxia-associated retinal neovascularization with leaky neovessels with increased permeability, hemorrhage, retinal detachment and consequent vision loss, as the disease further progresses into diabetic macular edema (DME) and proliferative DR, the advanced stages of DR [49,197,198].

Many factors contribute to DR pathophysiology, including loss of endothelial TJs, increased VEGF levels, oxidative stress, inflammation, sustained hyperglycemia-led advanced glycation end (AGE) products accumulation, and pericyte apoptosis resulting in increased vascular permeability [198,199,200]. In addition, DR also involves compromised oBRB owing to loss of RPE TJs, and RPE apoptosis [201,202,203,204]. Several treatment modalities for DR and DME includes anti-VEGF or steroid intravitreal injections, and laser photocoagulation [205,206,207,208]. Much of our understanding of DR pathologies comes from a broad range of diabetic animal models [209], including those with genetic manipulation, modified diet, pharmacological agents, and surgical procedures, to mimic different types and features of DR [210].

#### 3.4.1. Streptozotocin-Induced Type I Diabetic Mouse

Streptozotocin (STZ), a DM-inducing chemical and toxin, destroys pancreatic β cells, causing hyperglycemia and hypoinsulinemia, modeling type 1 DM [211]. It can be administered at either multiple-low doses or a single-high dose for inducing β cell damage [212]. Eyes of STZ-induced diabetic mice exhibit retinal nerve fiber damage and retinal ganglion cell layer thinning at 6 weeks, that deteriorates progressively with age. These mice show early DR pathologies with microvascular changes including increased vascular permeability by the eighth day after STZ injection, with VEGF upregulation, loss of TJ proteins and subsequent BRB breakdown [213]. STZ-induced hyperglycemic mice develop acellular capillaries accompanied by loss of pericytes [213,214].

#### 3.4.2. Akita (Ins2^Akita^) Mice

The Akita mice are a monogenic type I diabetic mouse model with spontaneous missense mutation in the insulin 2 gene (*Ins2*) [215]. This *Ins2* mutation hinders proinsulin transport from endoplasmic reticulum (ER) to Golgi, causing mutant proinsulin accumulation with subsequent ER stress and β-cell apoptosis. Heterozygous Ins2 mice hence have dysfunctional pancreatic β-cells without obesity or insulitis and develop hyper-glycemia and hypo-insulinemia by 4–6 weeks old [215,216].

*Ins2^Akita^* mice mimic type 1 DM with some but not all signs of DR [216]. Male heterozygous mice show retinal vascular abnormalities such as pericyte ghosts, vessel leakage and microaneurysm by 6 months of age, and pathological intraretinal neo-vessel formation in the outer plexiform layer by 9 months of age, but importantly without preretinal neovascularization as featured in proliferative DR [215,216]. Disorganized occludin immunoreactivity contributes to increased paracellular permeability and hence BRB dysfunction in these mice, along with vascular inflammation. In addition, visual function is impaired with severely attenuated ERG *b*-waves and oscillatory potential along with impaired synaptic connectivity and cone degeneration at 9 months age [216,217]. Together with glia and neuronal changes they have served as a mouse model for studying diabetic retinal neuropathy [218].

Despite the limiting concerns with this model lacking the advanced DR pathologies such as neuronal or capillary loss, retinal fluid exudation and preretinal neovascularization, *Ins2^Akita^* mice have been useful in exploring DR-related retinal complications in combination with other genetic factors or modifications in key immune factors [219,220]. For instance, elevated levels of IL-17, a cytokine from Th17 cells, have been associated with DR [220]. Mice deficient in Interferon-γ (GKO) are unable to promote Th1 immune response, hence promoting differentiation of Th cells into Th2 and Th17 and their activation. A double mutation *Ins2^Akita^GKO* mice, generated by crossing Akita and GKO, showed increased retinal VEGF and ICAM-1 levels, retinal exudates visible as hyperfluorescent areas and retinal capillary leukostasis [220], suggesting involvement of Th17 cells in DR pathogenesis and vascular leakage.

#### 3.4.3. Kimba (VEGF^+/+^) and Akimba (Ins2^Akita^VEGF*^+/^^−^*) Mice

Kimba, a low hVEGF expressing transgenic mouse strain (trVEGF029), transiently overexpresses *hVEGF* in photoreceptors as it contains hVEGF165 driven by a truncated rhodopsin promoter [179]. These mice develop retinal venous anomalies like tortuous vein and dilated capillaries by P28, with increased vascular permeability, and hyperfluorescent structures resembling microaneurysms, focal regions of neovascularization in INL and ONL, and retinal thinning 6 weeks after birth. Hence, this model mimics vascular changes as seen in non-proliferative DR or early proliferative DR stages. However, despite resembling the DR-associated vascular pathology, it lacks the hyperglycemic aspect of DR background [69,179,219,221].

To explore the roles of VEGF upregulation and hyperglycemia, two key factors in DR, a crossbreed of Akita and Kimba mice, the Akimba mouse (*Ins2^Akita^VEGF^+/^^−^*), was developed [69]. Akimba mice showed higher blood glucose levels at a younger age than Akita mice and pre-retinal neovascularization, resembling clinical signs of advanced DR. These mice manifest enhanced age-associated retinal changes, such as photoreceptor loss, retinal thinning, and more severe retinal abnormalities including vessel constriction, neovascularization, BRB loss, increased vascular leakage, and edema than Kimba mice. BRB disruption seen in these mice is associated with focal pathological angiogenesis and increased PLVAP expression. Increased retinal vascular permeability is a result of EC TJ protein downregulation and pericyte loss, hence posing as an excellent model for DR and DME [69,210].

#### 3.4.4. Non-Obese Diabetic (NOD) Mice

NOD mice, another spontaneous type 1 DM model, has a mutation in the CTLA-4 gene encoding an immune checkpoint regulatory protein. They hence have genetic predisposition to autoimmune diabetes and develop hyperglycemia due to targeted disruption of pancreatic β cells via CD4+ and CD8+ -mediated autoimmune response [222,223]. These mice show spontaneous onset of glucosuria, hyperglycemia with sudden weight loss, loss of pancreatic β cells and reduced size of islets resulting in insulin deficiency. NOD mice show systemic pathophysiology resembling human DM, such as development of autoantibodies, increased circulating autoreactive T cell levels prior to disease onset, which is accompanied with progressive loss of β cells [222,224]. These mice develop hyperglycemia by 12 weeks of age and exhibits retinal vascular damage [225]. Subsequent retinal vascular pathologies include capillary basement membrane thickening and blood vessel narrowing at 1-month post hyperglycemia, and retinal vascular degeneration and neovessel formation at 4-months post hyperglycemia [226]. Modified NOD mice with intravitreal pro-inflammatory cytokine injections are characterized by hyperglycemia and inflammation [227]. NOD mice have been primarily used for autoimmune diabetes study, however, DM development in these mice exhibit strong sex-specific prevalence with earlier onset (10–14 weeks old) and higher incidence (80%) in females in contrast to late onset (20 weeks old) and lower incidence (20%) in males, posting a limitation to this model [223,226].

#### 3.4.5. Leptin-Deficient (*Lep^ob/ob^*) Diabetic Mice

Leptin hormone is primarily involved in glucose and lipid metabolism, energy homeostasis and immunity by influencing appetite and satiety through its receptor (*ObR*). Produced by mature adipocytes, leptin also stimulates proliferation and angiogenesis of ECs and promotes VEGF-coordinated ischemia-induced neovascularization. This function of leptin is exerted through stimulating release of angiogenic factors, such as VEGF and fibroblast growth factor (FGF)-2, to induce EC proliferation as well as vascular progenitor cell mobilization, and vascular permeability [228]. Defective leptin signaling therefore contributes to pathological retinal vascularization and is related to metabolic diseases, like type 2 DM, obesity [229,230].

Leptin-deficient (*Lep^ob/ob^)* mice that arose from a spontaneous mutation in leptin gene mimic type 2 DM and are recognizable by 4 weeks of age. Mice with homozygous mutation are sterile [231]. *Lep^ob/ob^* mice with C57BL/J background exhibit mild hyperglycemia by 8–12 weeks, while on C57BL/KsJ background this mutation resulted in sustained hyperglycemia and diabetes, accompanied with pancreatic β-cell dysfunction [232]. Retinal changes associated with leptin deficiency include retinal stress, inflammation, reduced ERG *a*- and *b*-wave [233]. BTBR *ob/ob* mouse model exhibit retinal inflammation, early visual dysfunction, and vascular abnormalities, such as EC loss, primary plexus vessel degeneration, increased vessel permeability with exudation, and signs of BRB breakdown [234,235].

#### 3.4.6. Leptin Receptor-Deficient (*LepR^db/db^*) Diabetic Mice

Diabetes (db) mutation in *LepR* gene was first found in C57BL/KsJ strain, with homozygous mice showing severe DM signs with early disease onset, obesity by 3–4 weeks and subsequent hyperglycemia, glycosuria, polyuria, and retinal ganglion cell death [236]. The *db/db* mice have spontaneous mutations in leptin receptor and mimics the pathophysiology of human type 2 DM including obesity and late stage β cell atrophy [230,236]. Although both *ob/ob* and *db/db* mice models human type 2 DM, the *db/db* mice manifest significantly more severe obesity and diabetes, whereas *ob/ob* mice have diabetes with mild insulin resistance [230].

Retinal abnormalities in *db/db* mice include progressive retinal thinning observed at 16 and 24 weeks of age accompanied with mild and significant attenuation of ERG *a*- and *b*-wave amplitude, respectively, at 16- and 24-weeks age [237]. Diabetic mice retina show loss of pericytes and ECs, formation of acellular capillaries, and retinal vascular leakage [204,238]. Hence these leptin receptor-deficient mice are suitable models for investigating type 2 DM-associated retinal vascular pathology and dysfunction [239].

### 3.5. Mouse Model of Oxygen-Induced-Retinopathy (OIR)

In addition to DR, another vasoproliferative retinal disease characterized by neovascularization progressing to retinal detachment and vision loss is retinopathy of prematurity (ROP), prevalent in premature neonates. As described earlier, human retinal vasculature starts developing from central region by GW 16, toward periphery with complete vascularization just before birth in full-term neonates. However, preterm neonates have partially developed and incomplete retinal vasculature at birth, with normal retinal vessel growth further suppressed by the relative hyperoxic extrauterine environment and routine supplemental oxygen provided to these infants [240]. As the infants grow and the retinal neurons mature with higher metabolic demand, the incomplete vasculature sets the stage for tissue ischemia and subsequent hypoxia that induces abnormal vessel proliferation. Hence ROP progresses in two distinct phases: an initial phase of incomplete vessel growth due to prematurity that progresses to abnormal hypoxia-induced pathological vessel proliferation, with the late proliferative features somewhat similar to other proliferative retinal diseases, such as PDR [240]. BRB breakdown due to high levels of hypoxia-induced VEGF and extensive leakage from the abnormal neovessels contribute to the widespread vascular leakage in ROP [36]. Unlike humans, in mice, development of the retinal vessels starts right after birth and progresses rapidly in a regulated manner within a couple weeks; this species difference provides the advantage of investigation of retinal vasculature growth and patterning with the ease of genetic manipulation in mice.

The mouse OIR model reported in 1994 [241] represents a commonly used animal model of ROP that replicates both phases of ROP, with consistent and robust reproducibility of retinal neovascularization that can be quantified [242]. Development of this model involves exposure of neonatal mice at P7 to 75% ± 2% of oxygen for 5 days (P7–P12; phase I), followed by room air exposure for consecutive 5 days (P12–P17; phase II). Exposure to hyperoxia during phase I causes retinal vessel regression resulting in vaso-obliteration, imitating the initial phase of ROP. The second (proliferative) phase of human ROP is mimicked by hypoxia-induced gradual revascularization and pathological neovascularization on the border of the vaso-obliterated areas, upon return to the room air. OIR mouse model mirrors both key features of ROP, vaso-obliteration, and neovascularization, within a span of 2–3 weeks, with maximal retinal neovascularization at P17, that regresses spontaneously afterwards by P25. OIR-exposed mice exhibit increased retinal vessel leakage with significantly reduced expression of claudin-5 and occludin [243]. Besides dysfunctional BRB, OIR-exposed mice also show reduced inner retinal thickness due to delayed neuronal development, with significantly attenuated amplitude of ERG *a*- and *b*-waves at 4 weeks, decreased oscillatory potential amplitude, and increased Müller cell gliosis, indicating both morphological and functional abnormalities [243,244]. Additionally, this neovascular phase of mouse OIR model also resembles, to some extent, the PDR pathophysiology, albeit in the absence of hyperglycemia [240].

### 3.6. Mouse Models of Retinal Artery/Vein Occlusion (RAO/RVO) Exhibit Features of Vascular Leakage

Retinal vein occlusion (RVO), characterized by thrombotic obstruction of the retinal veins, results in retinal ischemia and hypoxia, and upregulation of hypoxia-induced growth factors that trigger abnormal neovascularization, vascular leakage, and macular edema leading to impaired vision [10,245,246]. Similarly, retinal artery occlusion (RAO) involves inner retinal infarction caused by blood clot formation in the artery [247].

Experimental mouse models exhibiting branched RVO have been developed that reproduce occlusion in retinal vein/artery to achieve certain features of vascular leakage. These were carried out with several similar approaches, like laser photocoagulation with photosensitizers [248,249], chemical-induced vasoconstriction, clamp-/suture-based occlusion [247], and diathermic cauterization [246]. The laser photocoagulation-based models [249] however, have a few limitations, such as absence of cystoid edema (often seen in INL) and retinal nonperfusion, both major features of RVO; and laser-induced inflammatory and photoreceptor damage in irradiated regions [248]. A more effective laser-induced RVO mouse model using ddY (Deutschland, Denken, Yoken) mice, a model of postprandial hyperlipidemia, has been reported [248], exhibiting swelling and cystoid edema in nerve fiber layer, INL and outer plexiform layer and increased INL thickness one day post 532 nm laser-guided photocoagulation. These mice showed remarkably reduced *a*- and *b*-wave ERG amplitudes representative of reduced visual function, and increased VEGF and ICAM-1 expression, where anti-VEGF treatment ameliorated the pathological effects [248]. In the same study, cystoid edema development was not seen in C57BL/6J or BALB/c mice under the same conditions, indicating strain dependence in model establishment.

## 4. Assessment of BRB Leakage

With the establishment of preclinical animal models of iBRB impairment, visualization of the retinal vascular leakage and quantification of its severity require robust and reliable methods to assess changes in retinal vascular permeability. Many in vivo and in vitro methods have been utilized to investigate the mechanisms underlying compromised ocular barrier integrity and evaluate the efficacy of potential therapeutic drugs. While this review focuses on assays of vascular barrier breakdown in the eye, for a comprehensive description of angiogenic bioassays in general, please see a review by Nowak-Sliwinska et al. [250].

### 4.1. In Vivo Methods to Assess BRB Leakage

#### 4.1.1. Fundus Fluorescein Angiography (FFA) 

Organic anions like fluorescein dye can easily diffuse out of leaky retinal/brain vessels and absorbed by the vitreous gel/brain parenchyma upon BRB/BBB breakdown in pathological conditions [251]. One well-established in vivo imaging technique is FFA [252], which uses intravenous or intraperitoneal injection of sodium fluorescein (NaFluo) tracer dye to image the retinal and/or choroidal vascular circulation. NaFluo is of low molecular weight (376 Da), inert, freely diffusible, and water soluble; these are its main advantages. However, its signal has poor penetration through RPE, because both its absorption (blue, 465–490 nm) and emission (green, 520–530 nm) wavelengths are in the visible light spectrum, hence limiting its use to image the choroidal circulation [253]. For normal choroid, indocyanine green angiography (ICGA) with tricarbocyanial dye in the infrared spectrum offers better RPE penetration and imaging capacity [254,255,256]. While some fraction of injected NaFluo dye binds to plasma proteins particularly albumin, a considerable portion of the tracer remains unbound in blood circulation [257]. Intact mature CNS capillaries does not leak fluorescein, however, new or diseased brain and retinal/choroid vessels or RPE with incomplete or impaired barrier permits leakage of tracer into the extracellular space [257], with its extravasation indicating compromised microvascular permeability to small molecules (Figure 3A).

FFA has been extensively used as a diagnostic and research tool in clinical and experimental settings [251,258,259]. The small molecular weight and high degree of fluorescence allow the NaFluo dye to pass more readily across barriers than large molecular weight tracers, hence serving as a rapid and sensitive indicator of impaired CNS barriers for early detection of localized vascular leakage and sites of retinal vein occlusion [253,260]. In addition, this technique is cost effective and non-toxic as the dye is eliminated in urine within 24–36 h of administration [261]. However, despite its many advantages, quantification of conventional fluorescein images is limited by the lack of accurate timing, reproducibility between studies and limited information regarding precise leakage location due to lack of depth resolution [253,260]. Another technical limitation is the restricted retinal field of view. Conventional FFA explores 30°–50° of retina in a single shot, restricting the visualization of peripheral retina that is crucial to retinal vascular studies [262]. Current wide-field and ultra-wide-field retinal imaging expands the single imaging surface up to >30°–<200° and 200°, respectively, and are being widely used for retinal diseases such as DR [263]. Compared with FFA which is optimal for leakage superficial to RPE, an infra-red fluorescent tracer, ICG, is better for examining choroidal blood flow or detecting pathologies like ischemia, neovascularization, and subretinal inflammation [264]. Nevertheless, ICG also has few disadvantages, such as high tissue uptake plus retention, non-specific binding resulting in substantial background scatter and gastrointestinal toxicity [265].

#### 4.1.2. Vitreous Fluorophotometry (VF)

VF is another commonly used procedure also based on NaFluo dye. A fluorometer attached to a slit lamp scans the vitreous cavity to measure fluorophore concentration in the anterior chamber, vitreous, choroid/retina, leaked from BRB [266]. Fluorometers are useful for many ocular applications, including drug pharmacokinetics, aqueous flow, and vascular leakage particularly for DR studies in humans [267], mice [268] and rats [269,270]. Recently, VF has been used to examine real-time measurement of vitreous and retinal permeability in mouse models of ocular disease following anti-VEGF treatment [266]. A few advantages of VF include its accuracy, quantitative nature, and ability to monitor permeability longitudinally in the same individual/animal hence allows evaluation of pre- and post-drug effects, just like FFA. It can be combined with other imaging procedures, such as FFA, to allow both spatial imaging and quantitative readout [266,271]. Limitations of VF readings include its inability to separate fluorescence signals from the choroid and retina, and hence impact of mixed choroidal and lens signals on retinal fluorescence [272,273].

#### 4.1.3. Fluorescent Microsphere Beads with Ophthalmoscopy

Real-time assessment of hypervascular permeability of retinal vessels can be performed using intravenous injection of fluorescent dye-conjugated polystyrene microspheres followed by confocal scanning ophthalmoscopy to scan the retinal surface [274]. These fluorophores emit in either the visible or near-infrared spectra. The deeper penetration of infrared wavelength makes near-infrared fluorophores (NIRF) the preferred choice to visualize deeper ocular tissues such as choroid. Also, NIR region offers reduced photo scattering and photo absorption by water and hemoglobin, hence low tissue autofluorescence [275]. Despite this, some NIRF agents have toxic effects due to organ retention and produce poor contrast images [276], however, net zero charge NIRFs such as ZW800, ZW700 reportedly produce superior quality images and undergoes rapid renal clearance with much lower toxicity [265,277]. Recently, these microsphere beads have been promoted as a non-invasive tool for in vivo imaging of choroidal/retinal vascular leakage in mice [278,279,280].

#### 4.1.4. Optical Coherence Tomography (OCT)

OCT is a non-invasive and high-resolution imaging modality widely used in clinical ophthalmology. It provides cross-sectional and 3D retinal images with depth-resolved reflectivity information to enable detection of structural abnormalities, such as retinal thickness, macular edema, retinal layer disruption/detachment [281,282,283]. Structural OCT can monitor and detect vascular fluid leakage and quantify macular volume or non-cystic macular thickening indicative of active vascular leakage for instance, in uveitis [284,285], however, it is unable to directly image the retinal or choroidal vessels [286]. OCT can detect inflamed retinal vessels identified by the hyperreflective vessel walls/lumen, enlarged vessels, thickened perivascular retina, and presence of vitreous inflammatory material in ocular diseases [287,288]. A combinational approach using FFA and spectral domain OCT is more sensitive to detect BRB alterations and sites of fluorescein leakage in DR patients [251].

Importantly, recently developed OCT angiography (OCTA) allows dye-free visualization of both retinal and choroidal vasculatures at micron-scale resolution using motion contrast of flowing erythrocytes in capillaries [289,290]. Compared with FA images of mostly the superficial vascular plexus, OCTA offers visualization of the intermediate and deep capillary plexus and also the radial peripapillary capillary network, however, without identification of structure BRB loss [291,292,293]. Recently, OCT microangiography has been established that captures sequential B-scans of the same location and provides 3D information of retinal and choroidal vessel morphology. It can identify neovessels, quantify retinal capillary dropout, characterize foveal avascular zone and parafoveal capillary plexus, yet cannot pinpoint leakage sites [289,294,295].

### 4.2. Ex Vivo Methods to Assess iBRB Leakage

In vivo techniques such as FFA, OCT, and VF are reliable and have contributed immensely to our understanding of retinal vascular disease pathogenesis and diagnosis. Yet these methods depend on access to specialized instruments which may not be universally available. More readily accessible quantitative ex vivo imaging and assays of retinal/choroidal vessel permeability are needed for evaluation of molecular regulators of BRB on experimental animal models. These primarily involve the use of easily diffusible, small endogenous or exogenous tracer molecules as discussed below, which allow the possibility of high-resolution microscopic imaging of whole retinas and quantitative analysis of vascular leakage into the retinal space.

#### 4.2.1. Colorimetric and Fluorometric Assays

Endogenous and exogenous colorimetric or fluorescent markers or tracers can be evaluated by absorbance or fluorescent spectrophotometry, and by ex vivo imaging of tracers in retinal flat mounts. As endogenous protein markers are localized in situ with natural preservation of barrier, highly concentrated exogenous tracers can provide much higher imaging and quantitative sensitivity. However, upon leakage, endogenous proteins continue to diffuse and are therefore less reliable to quantify duration or progression of leakage.

*Fluorescein isothiocyanate-conjugated-dextran (FITC-dextran)*, a commonly used tracer in experimental animal studies, is composed of glucose polymer dextran-conjugated with FITC at multiple sites. Dextran is available in various molecular weights, ranging from 4–70 kDa, and hence advantageous for size-selective leakage studies of ion and solute (low molecular weight) and protein (high molecular weight) [296] to determine severity of inner [297,298,299] or RPE/outer BRB leakage [300]. As an exogenous tracer, FITC-dextran is ideal for assessing the exact onset and time course of barrier disruption. Additionally, the high sensitivity of fluorescein offers detection of low levels of extravasation [298], and serves as a valuable tool to assess BRB/BBB breakdown and vascular leakage in ocular/neurological pathologies [258,301,302,303] (Figure 3B). In addition to FITC-dextran, FITC conjugated albumin can also serve as a high molecular weight protein tracer to assess vascular permeability.

*Evans blue (EB)* dye, a tetrasodium diazo salt, is a low molecular weight (961 Da), lipophobic, inert tracer commonly used in CNS barrier research. Injected Evans blue binds to albumin (69 kDa) at a molar ratio of approximately 10:1 to form a high molecular weight protein tracer [304,305,306]. Under normal conditions, the albumin-bound EB dye is restricted within the blood vessels, whereas in pathological conditions with increased vascular permeability, it starts leaking into the surrounding tissues. Tissues with permeable vessels rapidly exhibit bluish color [257,307]. EB can be administered intravenously or intraperitoneally with comparable efficacy [308]. After clearance of excess dye from circulation, the extravasated dye in target tissues is extracted using solvents such as formamide and measured by visible or fluorescence spectrophotometry (Figure 3C) [309], with fluorescence measurement being more sensitive [310], although tissue autofluorescence may interfere and require background correction [311]. Additionally, whole-body in vivo imaging of the EB-injected animals [312] or fluorescence microscopy of retinal flat mounts [313,314] can aid visualization of vascular leakage in distinct anatomic locations.

EB has been extensively used in vascular research in the eye and the brain due to its simplicity, safety, cost effectiveness, and visibility of vascular leakage by naked eyes from multiple tissues [301,309,314]. However, this method has some drawbacks including lack of protein specificity in plasma albumin binding as it also binds to globulins at higher concentrations, the possibility of presence of free/unbound dye in animal, and possible binding with tissues. These limitations altogether may affect the quantification and potential in vivo toxicity [311]. Also, solution of EB dye in physiological saline has been reported to affect dye structure [311], potentially affecting its stability.

#### 4.2.2. Extravasated Serum Protein Immunostaining

As use of endogenous markers reflects physiological conditions as a crucial advantage over exogenous tracers, immunolocalization of these molecules, such as IgG, albumin, and fibrinogen, offers many benefits including detection by light or electron microscopy [311,315]. While, due to their large size, these protein markers cannot detect sites of minor leakage for small molecules, they are useful in detecting hemorrhage sites and hard exudates that consists of accumulated plasma proteins [311].

#### 4.2.3. Horseradish Peroxidase (HRP) Staining

One enzymatic tracer molecule frequently used in vascular permeability studies since 1960s is HRP [316,317]. Tissue section containing injected HRP in blood vessel lumen reacts with 3,3′-diaminobenzidine (DAB) forming a dark brown precipitate, revealing the BRB/BBB disruption sites, which can be visualized macroscopically [257,317] or using bright-field or electron microscopy [318]. HRP is normally restricted from the brain or retinal vasculature due to the presence of EC tight junctions and lack of vesicular transport. However, in pathological conditions BRB/BBB becomes permeable to HRP, indicative of barrier leakage as reported in several vascular pathologies [65,92]. TJs appear as electron-dense junctional plaque under electron microscope (Figure 4A). Ultrastructural location of HRP-filled transcytosed vesicle (Figure 4B) or leaky TJs with diffusive HRP can be observed in vascular endothelium under electron microscope.

### 4.3. In Vitro Methods to Evaluate BRB Leakage

Although the above-mentioned techniques help visualize and quantify vascular permeability and are indispensable in studying the molecular transport across CNS/retinal barriers, they often cannot differentiate between paracellular or transendothelial transport. Moreover, molecular composition of human and mouse CNS barriers have distinct differences [319], that mouse models may not fully represent human conditions. Exploration of molecular regulators of BRB hence necessitates in vitro cell-based procedures, for investigating the molecular basis of barrier development and to enable functional studies. Cell-based models are particularly useful in recapitulating macular diseases, such as AMD, where in vivo mouse models do not fully reproduce human pathologies. In addition, in vitro cell-based assays are rapid, robust, and offer high throughput evaluation while avoiding the cost of animal studies and animal variation. Commonly used assays include trans-endothelial electrical resistance (TEER) [320], and vascular leakage assays using low molecular weight fluorescent tracers [321] in brain/retinal endothelial cells or co-culture of multiple cell types to mimic in vivo conditions, whereas significant advances have been made to develop in vitro models that better emulate the physiological conditions.

#### 4.3.1. Trans-Endothelial Electrical Resistance

TEER assesses the integrity of TJs in endothelial and epithelial cell monolayers [322] by measuring electrical resistance or ionic conductance across cell monolayer (Figure 5A). Factors influencing TEER across a monolayer include cell morphology, tightness between intercellular connections, and membrane robustness [323,324]. Since its introduction in 1980s, it has become a widely approved method to determine integrity and permeability of BBB/BRB, due to its sensitivity and reliability [325].

TEER may monitor barrier integrity in live cells at different growth/differentiation stages as validation of proper barrier formation in confluency, or pre- or post drug treatments. TEER measurement can be used in evaluation of signaling molecules such as VEGF, a potent inducer of vessel permeability [326] or Wnt signaling ligands: Wnt3a and Norrin [14]. VEGF increases vascular permeability by altering transcellular [327] and paracellular transport through ZO-1, occludin phosphorylation, Cln5 disruption [51,197] and hence, is used frequently for drug-based barrier permeability studies [328,329,330]. Nevertheless, TEER values can be affected by several factors such as culture duration, medium composition, cell passage number, changes in junctional length, and temperature [41,320,331], which must be considered for data interpretation.

#### 4.3.2. Two-Dimensional (2D) In Vitro Models

Two-dimensional in vitro models, epitomized with transwell systems, were first used to mimic the human endothelial barriers. These systems primarily consist of human EC monolayer grown on top of a semipermeable membrane or ECs co-cultured with monolayer of supporting cells (pericytes and/or astrocytes) grown on either side of the transwell insert [332]. These systems allow for direct permeability measurement as well as account for the dynamic interactions among different cell types. With the use of angiogenic agents, such as VEGF, and/or angiogenic inhibitors or other modulators, these systems can be used to mimic and assess some features of pathological angiogenesis, characteristic of ocular diseases like DR or wet AMD or to determine the effects of harmful end products affecting iBRB [333,334,335].

Most of the established in vitro BRB models use primary vascular endothelial cells isolated from animals, or human donors to better mimic human diseases. Primary cell lines, such as human umbilical vein endothelial cells (HUVECs) or human retinal microvascular cells (HRMECs) are commonly used for developing in vitro iBRB systems. While HUVECs are widely accessible, they do not fully represent features of retinal vascular endothelium features. Immortalized cell line, such as ARPE-19 is often used to model oBRB in vitro [336], in addition to primary RPE cells. Recently, approaches using embryonic stem cells (ESCs) or induced pluripotent stem cells (iPSCs) have been developed to generate retinal neural cells [337], or vascular progenitor cells expressing EC and TJ markers [338]. Recently, a genetically modified hPSCs-derived reporter cell line has been used to screen EC barrier stabilizing compounds [328]. This CLN5-P2A-GFP reporter cell line consists of green fluorescent protein (GFP) inserted with CLN5 in hPSCs, which were further differentiated into vascular ECs to screen for CLN5-regulating compounds employing GFP expression as a surrogate measure of barrier integrity [328].

*EC transwell transcytosis assays*: While TEER measurement primarily represents the paracellular permeability, transwell transcytosis assays exploit the intrinsic transcytosis pathway [21], in particular clathrin- and caveolae-mediated routes, to assess transendothelial transportation. These in vitro assays depend upon the ability of ECs to transport tracer molecules from the apical to basolateral side and vice versa (Figure 5B). These assays require primary cells or cell lines either consisting of single cell type like ECs [339] or co-cultures [332,333] involving pericytes, astrocytes with ECs to mimic in vivo conditions. Cells are typically grown to confluency on permeable membranes to permit access to both the apical and basolateral chambers, followed by addition of desired tracer molecule with signaling agents and/or drug treatments. Depending on the type of tracer molecule, these assays can help determine the transcellular permeability across the monolayer. Use of radiolabeled markers such as ^14^C-sucrose [340], fluorescent probe-solute conjugates like FITC-dextran, serum protein conjugates like albumin [341], Evans blue dye-albumin conjugate [342], biotin-albumin [343], and enzymes like HRP [344] to assess barrier diffusion have been reported. In the last few years, a few studies have utilized fluorescent tracer conjugated transferrin [318,339] or enzymatic marker like HRP [92] based assays for evaluating BBB/BRB transcytotic permeability. For this purpose, extensive washing in both chambers is required after tracer loading to eliminate the influence from paracellular transport, before assessing tracers from transendothelial vesicles alone.

#### 4.3.3. Three-Dimensional (3D) In Vitro Models

Despite the prevalent use of 2D cell culture systems, they exhibit limited resemblance of in vivo physiology of human eyes, such as polarized expression of certain proteins or receptors [336]. Three-dimensional cell-based systems that overcome this limitation have gained much attention in the past few years.

*Organ-on-chip models*: Micro physiological organ-on-chips are advanced 3D culture systems, representing a powerful tool to emulate the vascular barriers, such as iBRB [345], oBRB [346] and BBB [347]. Organ-on-chip are miniature apparatuses with cells from certain tissues/organs grown inside a microfluidic device, usually consisting of multiple cell types assembled as 3D constructs in a controlled microenvironment. They are designed to provide tissue-specific architecture recapitulating the in vivo-like molecular, structural, and physical features, and thus considered superior models to conventional 2D systems [336]. These systems offer flexibility over the spatial configuration of cells and their microenvironment. Additionally, these co-cultures can be transplanted in vivo to evaluate efficacy of treatment agents or monitor disease progression [250]. Cells can either be placed in between two chambers or encased in a 3D matrix, with or without an artificial membrane. Hydrogels, in place of membranes, offers direct cell-ECM contact, useful to study cell–matrix interaction studies [336]. Because cell–matrix communication is crucial for BRB function and maintenance, various matrix materials, such as alginate-hydrogels, silk fibroin membranes, and basement membrane matrix, have been studied as EC/RPE substrates for BRB models [348,349]. Integration of physiological flow in these systems is another major advantage, as shear stress modulates gene expression and contributes to barrier properties [336,350]. Additionally, these microscale systems allow for TEER measurement using microelectrodes, and live high-resolution imaging with minimal reagents and cells, by offering media-to-cell ratio comparable to physiological values [350].

*Organoid tubular channels*: Three-dimensional-vascular organ-on-chip systems can be established by co-culturing different cell types in pre-designed hydrogel scaffolds, hence mimicking the in vivo retinal/brain microvasculature. These consists of EC monolayer coated microfluidic channels, forming a tube or vessel-like structure adjacent to a gel-based mixture of different cell types [250,351]. Such a system using human retinal microvascular cells has been developed to assess permeability responses using leakage mediators in a retinal microvasculature-on-a chip model [352]. Although these models are useful to study barrier permeability or TJ protein expression, but they still cannot fully epitomize the in vivo permeability or morphology, such as vessel diameter and branching [351].

*Retinal organoids with vascular components*: Recently, combination of 3D cell culture systems with stem cells have advanced new research, such as iPSC-based organoids [353] and 3D-bioprinted constructs [354,355]. Self-formation of optic cup-like retinal organoids using mouse and human stem cells can closely emulate the retinas along with apical-basal polarity and spontaneous self-assembly into laminated neural structures [356,357]. These organoids can be useful in studying retinal development and degeneration, gene therapy and drug screening [355,358]. Moreover, BBB organoids consisting of ECs, astrocytes, pericytes co-culture have demonstrated upregulated TJ components, leukocyte adhesion molecules and efflux pumps [359], suggesting the promise of this platform in future BBB/BRB studies. Recently, iPSCs-derived light-sensitive human retinal organoids have also been developed with functional synapses [360]. These organoid models offer many advantages, such as the ease of culture/assembly, cost-effective, scalability and can be used to screen small and large molecule permeability across barriers [351], and hence hold much potential for future research. On the other hand, retinal organoids are labor-intensive and have limited translational application due to donor-to-donor variability or high heterogeneity between different cell lines [351,358]. In addition, development of nonneuronal vascular cells such as retinal vascular endothelial and choroidal cells in retinal organoids remains a challenge.

## 5. Summary

BRB is essential for maintaining proper visual function and retinal health. Impairment of barrier property and resultant retinal vascular leakage can lead to retinal edema and vision loss in many eye diseases. Over the years, many animal models with iBRB dysfunction and breakdown have been established (Table 1) modeling different eye diseases in humans with vascular permeability, each of which has distinct vascular features. In addition, multiple research techniques to assess vascular leakage have been evaluated ranging from in vivo and ex vivo imaging to in vitro assays (Table 2), with newer development focusing on better quantification algorithms for retinal vascular imaging, and novel assays to allow detailed mechanistic exploration and to facilitate anti-permeability drug screening. Together these models and methodologies have been widely utilized to investigate disease pathogenesis and greatly advanced our current understanding of cellular and molecular regulators of BRB, including the roles of endothelium paracellular transport via tight junctions, and vesicle-mediated transcytosis, as well as signaling pathways such as Wnt signaling. This review provides an overview of currently available animal models and assay methods with discussion of their advantages and limitations, to assist researchers in this field choose their optimal experimental tools. Future advances in BRB studies will allow better treatment and management of vascular leakage to protect retinas and neurons, and for developing new ways to enable transient opening of barrier function and allow pharmacodelivery across barrier into retinas and CNS.

## Figures and Tables

**Figure 1 cells-12-02443-f001:**
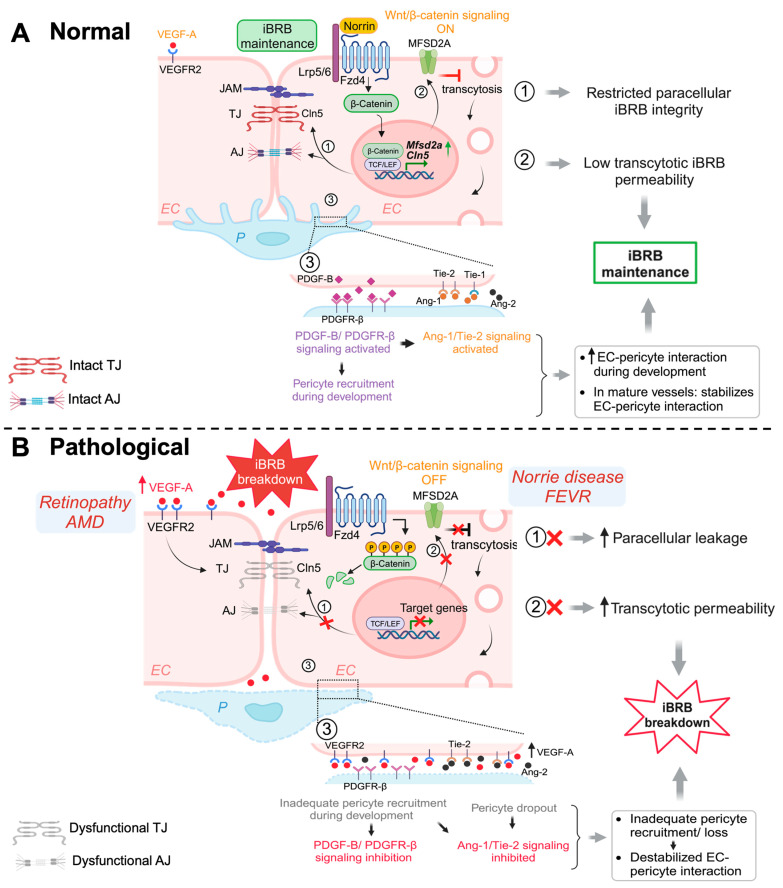
**Schematic illustration of multiple signaling pathways important for maintaining iBRB integrity.** (**A**) **Normal**: Activation of Wnt/β-catenin pathway during development and in adult retinal vessels contributes to the formation and maintenance of iBRB in the vascular endothelium. Under normal (ON) conditions, Wnt ligand (norrin) binds to Fzd4 receptor and Lrp5/6 co-receptor causing stabilization of cytosolic β-catenin, which then functions as a transcriptional signaling molecule by translocating to the nucleus and binding TCF/LEF to mediate target gene expression. Wnt/β-catenin signaling increases expression of TJ component gene *Cln5* to limit paracellular permeability and also restricts EC transcellular transcytosis by activating transcription of *Mfsd2a*, hence maintaining iBRB integrity. In addition, β-catenin also strengthens EC adhesion as a structural component of adhenrens junctions. Under normal conditions, physiological levels of VEGF through its receptor VEGFR2 signaling sustains and maintains normal vessels. Development and maintenance of iBRB also depends on EC interaction with mural cells including pericytes. During development, recruitment of pericytes by nascent endothelial tip cells occurs through PDGF-B/PDGFR-β signaling, which together with angiopoetin/Tie2/signaling cascade, promoting EC-pericyte interaction. Activation of Tie2/Tie1 signaling also maintains the iBRB integrity in intact adult retinal vessels. (**B**) **Pathological:** Under pathological conditions with genetic deficiency of Wnt signaling components, the absence of Wnt ligands (OFF state) results in phosphorylation and proteasomal degradation of cytosolic β-catenin, inhibiting expression of downstream genes such as Cln5 and Mfsd2a, thereby resulting in loss of iBRB and increased vascular leakage. In prevalent vascular eye diseases of proliferative retinopathies and neovascular AMD, increased levels of VEGF act mainly through VEGFR2 to impair TJ components and also affects transcytosis, leading to iBRB breakdown, vascular leakage, and retinal edema. Inadequate pericyte recruitment during development due to impaired PDGF-B signaling and altered Tie2 signaling can both lead to destabilized EC-pericyte interactions, with pericyte dropout being a hall mark of diabetic retinopathy, impairing iBRB. EC, endothelial cell; P, pericyte; Fzd4, Frizzled-4; Lrp5/6, LDL-receptor-related protein 5/6; TJ, tight junctions; AJ, adherens junctions; Cln5, claudin-5; MFSD2A, major facilitator superfamily domain containing 2A; TCF/LEF, T-cell factor/lymphoid enhancer factor; PDGF-B/PDGFR-β, platelet-derived growth factor B/PDGF-receptor-β; Tie-1/-2, tyrosine kinase receptor-1/2; Ang-1/-2, Angiopoetin-1/-2; VEGF-A, vascular endothelial growth factor A; VEGFR2, VEGF receptor 2; iBRB, inner blood–retinal barrier. Figure was created with BioRender.com.

**Figure 5 cells-12-02443-f005:**
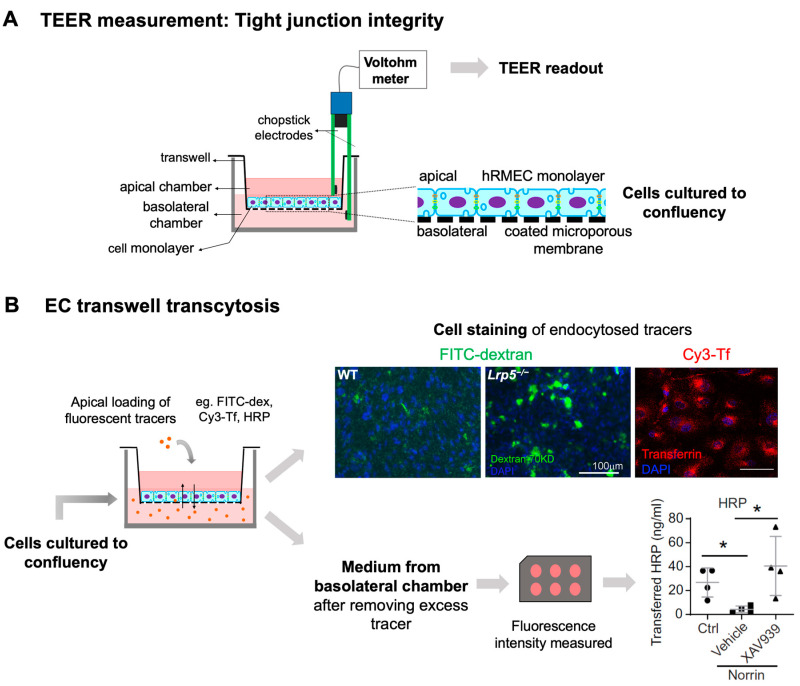
**In vitro methods for quantifying vascular leakage in cultured endothelial cells.** (**A**) A Schematic illustration showing transwell cell culture system and positioning of electrodes in the apical and basolateral chambers for TEER measurement of vascular permeability in cultured cell monolayer. Magnified region shows the structural arrangement of fully confluent endothelial cell monolayer on coated microporous membrane of the transwell. (**B**) In vitro fluorescent or enzymatic tracer-based transwell assays for assessing vascular leakage through paracellular transport and/or transcytosis. FITC-dextran-70 kDa quantifies vascular leakage through both transcellular and paracellular routes, whereas Cy3-Tf tracer and HRP can be used to assess vascular leakage through clathrin-mediated and caveolae-mediated EC transcytosis, respectively. FITC-dextran (green) or Cy3-Tf (red) endocytosed by HRMECs can be visualized using fluorescent microscopy. Vascular leakage across the EC monolayer can be quantified with fluorescence measurement of medium collected from the bottom chamber. TEER, trans-endothelial electrical resistance; FITC-dextran-70 kDa, fluorescein isothiocyanate conjugated dextran-70 kDa; EC, endothelial cell; Cy3-Tf, cyanin3-conjugated transferrin; HRP, horseradish peroxidase; HRMECs, human retinal microvascular cells. Scale bar: (**B**) 100 μm. * *p ≤* 0.05. Parts of the figure were adapted with permission from [318].

**Table 1 cells-12-02443-t001:** Mouse models of impaired iBRB and vascular leakage.

	Mouse Models	Genetic/Chemical Manipulation	Diseases Associated	Pathological iBRB Characteristics	References
**Defective Wnt/β-catenin signaling**	*Ndp^y/^* * ^−^ *	Loss-of-function mutation in norrin (*Ndp*) gene	X-linked Norrie disease and FEVR	Delayed superficial vascular layer development; absence of intermediate and deep retinal vessels; persistent hyaloid; microaneurysm-like vascular lesions; tortuous and fenestrated retinal vessels; diminished ERG *b*-wave; compromised iBRB/BBB.	[16,89,90,124,134,145]
	*Lrp5* * ^−/^ * * ^−^ *	Loss-of-function mutation in *Lrp5* gene	FEVR	Delayed superficial vascular layer development; absence of intermediate and deep retinal vessels; fenestrated retinal blood vessels; persistent hyaloid; microaneurysm-like vascular lesions; diminished ERG *b*-wave; compromised iBRB/BBB.	[56,85,126,131,142,144]
	*Fzd4* * ^−/^ * * ^−^ *	Loss-of-function mutation in *Fzd4* gene	FEVR	Delayed superficial vascular layer development; lack of intermediate and deep retinal vessels; failed hyaloid regression; enlarged, tortuous retinal vessels with fenestrations; progressive cerebellum degeneration, compromised iBRB/BBB.	[16,125,134,141,154]
	*Tspan12* * ^−/^ * * ^−^ *	Loss-of-function mutation in *Tspan12* gene	FEVR	Delayed hyaloid vessel regression; lack of intermediated and deep retinal vessels; fenestrated retinal vessels; glomeruloid vascular malformations; diminished ERG *b*-wave.	[87,127,147]
	*Tspan12^ECKO^*	EC-specific *Tspan12* inactivation	FEVR	Lack of intra-retinal capillaries; glomeruloid vascular malformations; retinal vascular leakage; impaired ERG *b*-wave; reduced mural cell coverage.	[87,146]
	*Ilk^iECKO^*	Inducible EC-specific *Ilk* knockout	FEVR	Reduced retinal vessel density and branching; compromised retinal sprouting with EC clusters; defective deep vessel layer; reduced EC proliferation; defective vascular remodeling.	[150]
	*Ctnna1^iECKO^*	Inducible EC-specific *Ctnna1* knockout	FEVR	Loss of iBRB/BBB integrity with bleeding regions in eye and cerebellum; slower hyaloid regression; defective vascular sprouting; absence of intermediate and deep vessels; leaky vessels, discontinuous EC TJs.	[129]
**Spontaneous retinal/choroidal neovascularization**	*Vldlr* * ^−/^ * * ^−^ *	Loss-of-function mutation in *Vldlr* gene	RAP	Abnormal vessel sprouting into subretinal space, intraretinal angiogenesis and choroidal anastomosis; subretinal hemorrhage, retinal edema, RPE disruption; increased retinal vascular leakage.	[168,361,362]
	*VEGF_164_/Rho*	Photoreceptor-specific hVEGF-overexpression	Diabetic retinopathy	Intra- and sub-retinal neovascularization; retinal vascular leakage; photoreceptor degeneration, INL disorganization; proliferative retinopathy and retinal detachment.	[176,177]
	*VEGF_164_/RPE*	RPE-specific mVEGF-overexpression	Diabetic retinopathy	Intrachoroidal neovascularization and leakage; does not induce CNV.	[180,181]
	*JR5558* (*NRV2*)	spontaneous VEGFA-dependent neovascularization	RAP	Spontaneous intraretinal neovascularization into the sub-retinal space; leaky neovascular lesions in ONL and subretinal space; reduced rod and cone-ERG wave; retinal edema, focal photoreceptor death; depigmented fundus.	[183,185]
	*Ap1g1^fgt^*	spontaneous 6-bp in-frame deletion mutation	choroidal vascular anastomosis	Spontaneous choroidal neovascularization; vascular leakage throughout retina; multiple areas of retinal depigmentation.	[191]
**Diabetic retinopathy and diabetic macular edema**	*VEGF^+/+^* (*Kimba*)	Photoreceptor-specific hVEGF (trVEGF029) overexpression	RAP and DR	Dilated capillaries, tortuous veins; increased vascular permeability, microaneurysms; focal areas of neovascularization in INL and ONL.	[179,221]
	*Ins2^Akita^* (*Akita*)	spontaneous missense mutation in the insulin 2 gene	Type I diabetes mellitus	Retinal vessel leakage, pericyte ghosts, microaneurysm; increased iBRB paracellular permeability; pathological OPL neovascularization; attenuated ERG.	[215,216,217]
	*Ins2^Akita^VEGF^+/^^−^* (*Akimba*)	Akita x Kimba mice	Early DR and DME	Retinal vessel constriction, neovascularization; pericyte loss; TJ protein downregulation; increased vascular leakage and edema; retinal thinning, photoreceptor loss.	[210]
	*NOD*	Encodes T1D-risk MHC haplotype	Type I diabetes mellitus	Retinal vascular damage; neovascularization, inflammation; early onset, more frequent in females.	[222,223,226]
	STZ-induced	Streptozotocin-induced pancreatic β cell destruction	Type I diabetes mellitus	Progressive NFL damage and thinning; increased VEGF expression; loss of TJ proteins; increased vessel permeability; pericyte loss and acellular capillaries.	[212,213,363]
	*Lep^ob/ob^*	Spontaneous loss-of-function mutation in leptin (*Lep*) gene	Type II diabetes mellitus	Retinal stress, inflammation; reduced ERG *a*- and *b*-waves.	[231,232,233,234]
	*LepR^db/db^*	Spontaneous mutations in leptin receptor (*LepR*) gene	Type II diabetes mellitus	Retinal thinning at 16–24 weeks; mild ERG *a*-wave and significant *b*-wave attenuation; ECs and pericyte loss, acellular capillaries; retinal neovascularization; retinal vascular leakage.	[236,237,238,239]
**ROP**	OIR	Oxygen-induced retinopathy	ROP	Retinal vessel obliteration; abnormal neovascularization; iBRB impairment; increased vascular leakage; TJ loss; thinner inner retina (INL, IPL, ONL) at 4- and 8-weeks; significant ERG *a*- and *b*-wave attenuation; retinal stress.	[240,241,242,243,244]
**RVO**	Laser photocoagulation in ddY mice	532 nm laser-guided photocoagulation	RVO	Swelling and cystoid edema in NFL, INL, OPL; increased VEGF and ICAM-1 expression; ERG *a*- and *b*-waves attenuation.	[245,248]

Abbreviations: BRB: blood-retinal barrier; BBB: blood-brain barrier; CNV, choroidal neovascularization; DME, diabetic macular edema; DR, diabetic retinopathy; ERG, electroretinogram; FEVR, familial exudative vitreoretinopathy; ICAM-1, intercellular adhesion molecule 1, INL, inner nuclear layer; IPL: inner plexiform layer; NFL, nerve fiber layer; OIR, oxygen-induced retinopathy; ONL, outer nuclear layer; OPL, outer plexiform layer; RAP, retinal angiomatous proliferation; RGC, retinal ganglion cell; ROP, retinopathy of prematurity; RPE, retinal pigmented epithelium; RVO, retinal vein occlusion; TJ, tight junction; VEGF, vascular endothelial growth factor.

**Table 2 cells-12-02443-t002:** Methods and assays to assess iBRB leakage in vivo and in vitro.

	Methods	Contrasting Agents	Advantages	Limitations	References
**In vivo methods**	OCT	Light source	Non-invasive, high resolution, live 3D retinal structural imaging for structural abnormalities, inflammation and leakage; better imaging of choroid and sclera with accurate choroid thickness measurement (EDI-OCT); analysis of both deep and superficial tissues and wider scan area (SS-OCT); dye-free high contrast images of vessels (OCTA).	High cost; low axial resolution; cannot detect sites of vascular leakage.	[260,283,290,294,364,365]
	FFA	Sodium fluorescein or Indocyanine green dye	Real-time imaging; visualizes retinal or choroidal vascular leakage; early leakage and leakage sites detection; inert non-toxic dye; highly sensitive, rapid, cost-effective; suitable for longitudinal studies; wide-field imaging offers increased imaging field.	Invasive; leakage quantification difficult; no depth-resolution; time-sensitive. ICG: high tissue uptake and retention; high background.	[252,253,256,258,263,264]
	Vitreous fluorophotometry	Sodium fluorescein	Real-time quantitative evaluation of vascular leakage; suitable for longitudinal studies; non-toxic inert dye; highly sensitive; rapid, cost effective.	Invasive; cannot image retinal/choroidal blood flow; high background fluorescence.	[266,268,272,273]
	Fluorescent microsphere beads	Fluorescent polystyrene microspheres	Real-time assessment of retinal/choroidal vascular permeability; NIR fluorophores exhibit low light-scattering and low tissue autofluorescence.	Invasive; poor contrast images; toxicity due to tissue retention.	[274,277,278]
**Ex-vivo methods**	Evans blue extravasation	Evans blue dye	Dye binds to endogenous plasma albumin; visible to naked eyes; cost-effective, simple, non-toxic; colorimetric/fluorescent detection; vascular leakage can be detected/quantified by different methods.	Binds to other plasma proteins; presence of unbound dye affects quantification.	[305,309,311,366]
	FITC-dextran	Dextran-conjugated with FITC	Availability of dextran with different molecular weight (4–70 kDa) for size selective studies; exogenous tracers; suitable for time course studies; inert complex and highly sensitive.	Light-sensitive; exposure to fluids reduces fluorescence; terminal method.	[258,296,298,300,302]
	Endogenous tracer immunostaining	Endogenous serum proteins	Stains endogenous proteins as tracers by light/electron microscopy; detects native hemorrhage sites and exudates of plasma proteins.	Large sized tracers; cannot detect minor leakage; terminal method.	[311,315]
	Exogenous tracer staining	Horseradish peroxidase	Suitable for imaging sites of vascular leakage by light or electron microscopy; disrupted TJ or transcytosed vesicle can be accurately located.	Detection by electron microscope is time-consuming; potentially carcinogenic substrate.	[92,257,367]
**In-vitro methods**	TEER	Ionic conductance	Live monitoring of monolayer integrity; evaluates paracellular barrier integrity; useful for pre- or post drug permeability studies in 2D/3D systems	High variability due to several factors.	[320,329,331,368]
	2-D models	Fluorescent or enzymatic tracers	Mimics in vivo conditions; potential co-culture of multiple cell types; assess transcytotic barrier permeability.	Cannot replicate accurate in vivo physiological conditions.	[318,332,333,335,338,339]
	3-D models	TEER or tracers	Better emulation of in vivo conditions with 3D cell–cell interaction and physiological flow; controlled microenvironment; TEER measurement; potential of co-cultures/stem cell approaches for drug/compound screening.	Labor-intensive; high heterogeneity between cell lines.	[345,346,352,356,358]

Abbreviations: FITC, fluorescein isothiocyanate; NIR, near infra-red; OCT, optical coherence tomography; EDI-OCT, enhanced depth imaging OCT; SS-OCT, swept source OCT; OCTA, OCT angiography; TEER, trans-endothelial electrical resistance; TJ, tight junction; 2-D, two dimensional; 3-D, three dimensional.

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
