# Peer review of "Assessment of Inner Blood–Retinal Barrier: Animal Models and Methods"

_cells, 2023, doi:10.3390/cells12202443_

Round 1

Reviewer 1 Report

This review manuscript on experimental models of iBRB is an excellent piece of work. It is well-written, easy to understand, thorough and very up-to-date. It was a pleasure to read it. The Table at the end of the manuscript summarizes clearly all mouse models used for studies of damaged iBRB.

I could find only one little aspect to comment upon:

The reference list is extensive and up-to-dat. However, I was surprized that an equally well-designed review  by the group of Griffioen and Nowak-Sliwinska is missing. I thinki these two reviews are nicely complementary:

Nowak-Sliwinska et al (2018) 21:425-532: Consensus guidelines for the use and interpretation of angiogenesis assays (PMID 29766399).

Besides this small issue, I can only congratulate the authors with this fine review manuscript. I am sure it will be frequantly cited once it is published.

Reviewer 2 Report

The review article entitled "Assessment of inner blood retinal barrier: animal models and methods" and submitted for publication in the journal Cells is a well-structured, very complete and well-illustrated paper that will surely be one of the works used by researchers in the field when they want to have a global idea of the models and methods to use when studying the BRB.

Author Response

Thank you. Please see PDF.

Reviewer 3 Report

The BRB is essential for maintaining the retina as an immune-privileged site. More importantly, the bi-directional movement of molecules across the BRB is strictly controlled for proper visual function. Disruption of the BRB is associated with a variety of retinal diseases, including diabetic retinopathy, retinal vein occlusion, retinopathy of prematurity, ocular inflammatory diseases and retinal degenerative diseases. This review paper provides an overview of iBRB development, cellular and molecular composition and signaling pathways involved in modulating iBRB function, summarizes mouse models with impaired iBRB and vascular abnormalities, and describes in vivo, ex vivo and in vitro methods available for qualitative and quantitative assessment of iBRB leakage. The manuscript is well-organized and comprehensively written. The schematic illustrations and summary tables are clear and concise. This review is of interest to help researchers in this field choose appropriate and optimal tools to study the iBRB and monitor the efficacy of different therapeutic strategies.

The authors are requested to address a few minor issues:

1. The contribution of glial cells to the iBRB mentioned in 2.4.4 is relatively simple. Please describe different glial cell types and their regulatory roles in maintaining the integrity of the iBRB (e.g. VEGF and HIF pathways, GDNF, TGFβ, as well as Wnt signaling).

2. Please also include OIR which is an experimental model of retinopathy of prematurity.  

3. A mistake on line 1109, it should be “4.3.3. 3D in vitro models”. 
